# Distinct progenitor populations mediate regeneration in the zebrafish lateral line

Eric D Thomas[1,2], David W Raible[1,2,3]*

[1]Department of Biological Structure, University of Washington, Seattle, United States; [2]Graduate Program in Neuroscience, University of Washington, Seattle, United States; [3]Virginia Merrill Bloedel Hearing Research Center, University of Washington, Seattle, United States

**Abstract** Mechanosensory hair cells of the zebrafish lateral line regenerate rapidly following damage. These renewed hair cells arise from the proliferation of surrounding support cells, which undergo symmetric division to produce two hair cell daughters. Given the continued regenerative capacity of the lateral line, support cells presumably have the ability to replenish themselves. Utilizing novel transgenic lines, we identified support cell populations with distinct progenitor identities. These populations show differences in their ability to generate new hair cells during homeostasis and regeneration. Targeted ablation of support cells reduced the number of regenerated hair cells. Furthermore, progenitors regenerated after targeted support cell ablation in the absence of hair cell damage. We also determined that distinct support cell populations are independently regulated by Notch signaling. The existence of independent progenitor populations could provide flexibility for the continued generation of new hair cells under a variety of conditions throughout the life of the animal.
DOI: https://doi.org/10.7554/eLife.43736.001

*For correspondence:
draible@uw.edu

Competing interests: The authors declare that no competing interests exist.

## Introduction

The regenerative potential of a given tissue is dependent on the availability of progenitor cells that are able to functionally replace lost or damaged cells within that tissue. For instance, bulge cells in the hair follicle can repair the surrounding epidermis (*Rompolas and Greco, 2014*; *Hsu et al., 2014*), new intestinal epithelial cells arise from crypt cells (*Santos et al., 2018*; *Yousefi et al., 2017*), and horizontal and globose basal cells can regenerate cells in the olfactory epithelium (*Choi and Goldstein, 2018*; *Schwob et al., 2017*). Depletion of these progenitors can severely diminish the regenerative capacity of the tissue, and tissues that lack a progenitor pool altogether are unable to regenerate. To gain further insight into how different tissues regenerate, a greater understanding of the mechanisms that define and regulate progenitor function are needed.

The zebrafish lateral line system has long been recognized as an excellent model for studying regeneration. The sensory organ of the lateral line, the neuromast, is comprised of mechanosensory hair cells organized on the surface of the head and body (*Thomas et al., 2015*). Lateral line hair cells regenerate rapidly following damage, with the system returning to quiescence after regeneration is complete (*Harris et al., 2003*; *Hernández et al., 2007*; *Ma et al., 2008*). The surrounding nonsensory support cells serve as progenitors for new hair cells. This replenishment is proliferation-dependent and occurs symmetrically, with each progenitor dividing to give rise to two daughter hair cells (*Wibowo et al., 2011*; *Mackenzie and Raible, 2012*; *López-Schier and Hudspeth, 2006*; *Romero-Carvajal et al., 2015*). Three key observations of support cell behavior during regeneration suggest that different support cell populations may be differentially regulated in response to regeneration. First, the support cell proliferation that follows hair cell death occurs mainly in the dorsal and ventral compartments of the neuromast (*Romero-Carvajal et al., 2015*), indicating that progenitor identity

**eLife digest** Deep inside our ears, tiny specialized cells called hair cells constantly detect and relay sound and spatial information to our brain. Without them, we lose our sense of hearing and balance. Unfortunately, the number of hair cells drops with age or after exposure to loud noises, and there is no way for our body to replace them. This can lead to permanent hearing and balance problems.

Zebrafish rely on similar hair cells to sense their environment. In particular, clusters of hair cells make up the lateral line system, an organ that helps fish perceive vibration and pressure in the water. However, unlike us, zebrafish can quickly and completely regenerate their hair cells. When these get damaged, surrounding 'support cells' divide to form new hair cells, but the details of this process were still vague. For example, it was unclear whether all support cells could create new hair cells, or only a certain population. There was also little evidence to show that support cells could regenerate themselves.

To investigate, Thomas and Raible used a precise genetic tool called CRISPR to label subsets of support cells that differed in their gene expression. Then, these cells were followed over time to see what they would become. This highlighted three distinct populations that played separate roles when hair cells were regenerated.

The support cells located at the top and bottom of the lateral line organs (dorsal and ventral cells) made most of the new hair cells. The cells located at the front and back (anterior and posterior cells) made a few; and the cells around the edges (peripheral cells) did not make any. Further experiments then showed that all three types of support cell could transform to replenish the stock of dorsal and ventral support cells that make new hair cells.

By being able to label and track precise groups of support cells, researchers will be able to dissect exactly how hair cells are regenerated in fish. Armed with this knowledge, it may become possible to explore ways to encourage human support cells to replace damaged hair cells in our ears.

DOI: https://doi.org/10.7554/eLife.43736.002

is spatially regulated. The most peripheral support cells, often called mantle cells, do not proliferate in response to hair cell damage (*Ma et al., 2008*; *Romero-Carvajal et al., 2015*). Second, the regenerative capacity of the neuromast is not diminished over multiple regenerations (*Cruz et al., 2015*; *Pinto-Teixeira et al., 2015*), indicating that progenitor cells must also be replaced in addition to hair cells. Finally, in addition to regeneration in response to acute damage, lateral line hair cells undergo turnover and replacement under homeostatic conditions (*Cruz et al., 2015*; *Williams and Holder, 2000*). However, it remains unknown whether there are distinct support cell populations within the neuromast (e.g. hair cell progenitors and those that replenish progenitors), as well as how progenitor populations are regulated.

In this study, we have used CRISPR to generate novel transgenic lines in which distinct, spatially segregated populations of support cells are labeled in vivo. Fate mapping studies using these lines show that these populations are functionally distinct with respect to their ability to contribute new hair cells during homeostasis and to generate hair cells after damage. We also show that targeted ablation of one of these populations significantly reduces hair cell regeneration. Other fate mapping studies show that these support cell populations can replenish each other in the absence of hair cell damage. Finally, we show that Notch signaling differentially regulates these populations. These results demonstrate that there are a number of distinct progenitor populations within lateral line neuromasts that are independently regulated, providing flexibility for hair cell replacement under a variety of circumstances.

# Results

## Hair cell progenitors are replenished via proliferation of other support cells

Previous studies have shown that the majority of support cell proliferation occurs during the first twenty-four hours following hair cell death (*Ma et al., 2008*). We replicated this finding by administering a pulse of F-ara-EdU (EdU), which has been shown to be far less toxic than BrdU (*Neef and Luedtke, 2011*). The EdU pulse was administered for twenty-four hours following neomycin-induced hair cell ablation at 5 days post fertilization (five dpf) and neuromasts were imaged at seventy-two hours post treatment (72 hpt), the time at which regeneration is nearly complete (*Figure 1A*). In neomycin-treated larvae, roughly 78% of regenerated hair cells were EdU-positive, compared to 6% in mock-treated larvae (*Figure 1B–C*; p<0.0001). We noticed that at the same timepoint that 28% of EdU-positive cells remained support cells (*Figure 1E and B* arrowheads). We hypothesized that these cells may represent hair cell progenitors that had been replaced via proliferation. If so, then these EdU-positive cells should have the capacity to generate a new round of hair cells after subsequent damage. In order to test this, we subjected larvae to two rounds of hair cell ablation and regeneration. EdU was administered for 24 hr following the first ablation, and BrdU was administered for the same duration following the second ablation (*Figure 1F*). Of the 90 neuromasts analyzed, 14 retained EdU labeling. Within these 14 neuromasts, we observed hair cells after the second regeneration that were both EdU- and BrdU-positive (*Figure 1G–J*, arrowheads; 1.50 ± 1.45 hair cells per neuromast; mean ±SD), indicating that support cells that divide after the first ablation can in fact serve as hair cell progenitors after subsequent damage. However, we also observed

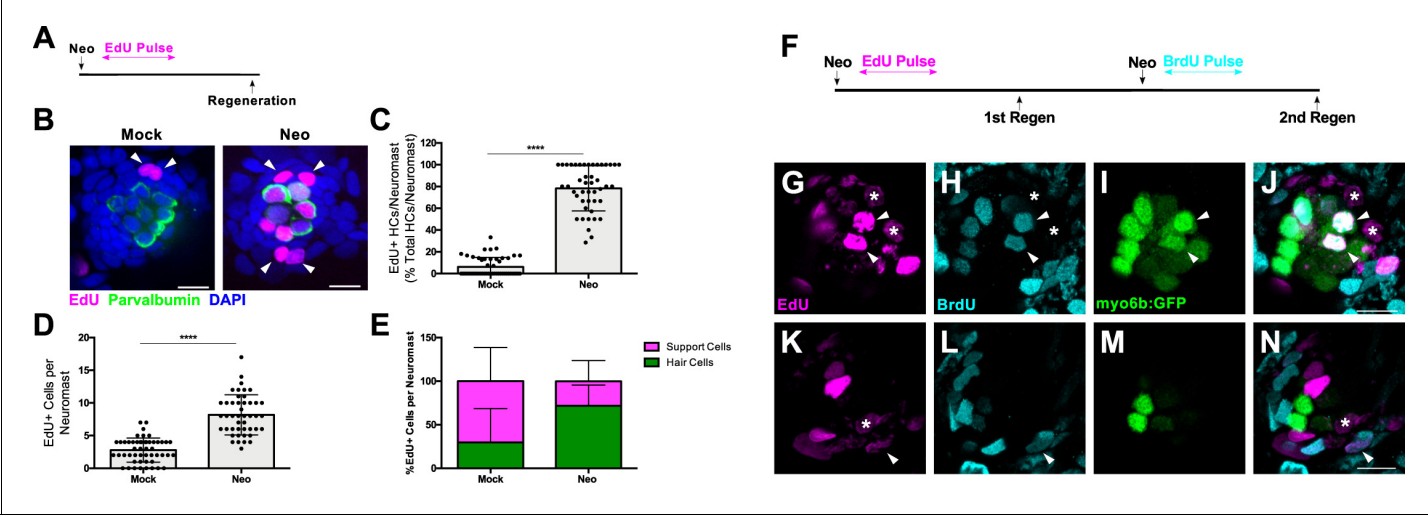

**Figure 1.** Hair cell progenitors are replenished via proliferation of other support cells. (A, F) Timelines of single-ablation (A) and double-ablation (F) proliferation experiments. (B) Maximum projections of mock- (Mock) and neomycin-treated (Neo) neuromasts. EdU-positive cells are shown in magenta, anti-Parvalbumin-stained hair cells are shown in green, and DAPI-stained nuclei are shown in blue. Arrowheads indicate EdU-positive support cells. Scale bar = 10 μm. (C) Percentage of hair cells per neuromast labeled by EdU. Mock: 6.11 ± 8.69, n = 50 neuromasts (10 fish); Neo: 78.24 ± 20.69, n = 45 neuromasts (nine fish); mean ± SD; Mann Whitney U test, p<0.0001. (D) Total EdU-positive cells per neuromast. Mock: 2.78 ± 1.84, n = 50 neuromasts (10 fish); Neo: 8.18 ± 3.07, n = 45 neuromasts (nine fish); mean ± SD; Mann Whitney U test, p<0.0001. (E) Percentage of EdU-positive cells per neuromast that are either hair cells or support cells. Mock: 29.73% hair cells, 70.27% support cells, n = 50 neuromasts (10 fish); Neo: 72.02% hair cells, 27.98% support cells, n = 45 neuromasts (nine fish); mean ± SD. (G–N) Individual slices of a neuromast following two regenerations at two different planes: apical hair cell layer (G–J) and basal support cell layer (K–N). EdU (visualized by a Click-iT reaction) is labeled in magenta, BrdU (anti-BrdU) is labeled in cyan, and myo6b:GFP hair cells are labeled in green. Arrowheads indicate EdU/BrdU-positive hair cells, and asterisks indicate EdU-positive support cells. Scale bar = 10 μm.

DOI: https://doi.org/10.7554/eLife.43736.003

The following source data is available for figure 1:

**Source data 1.** Hair cell progenitors are replenished via proliferation of other support cells
DOI: https://doi.org/10.7554/eLife.43736.004

double-positive cells that remained support cells (*Figure 1K–N*, arrowheads; 1.07 ± 0.73 support cells per neuromast; mean ±SD), as well as support cells that were only labeled by EdU (*Figure 1G–N*, asterisks). These observations indicate that support cells that divided after the first round of damage do not always serve as hair cell progenitors (or even as progenitors at all). Altogether, these data provide evidence of proliferation-mediated replenishment of hair cell progenitors by support cells in the neuromast, but also suggest that new hair cells arise from a pool of progenitors that are not strictly defined by their proliferation history.

## Different progenitor identities among distinct support cell populations

We next sought to determine whether hair cell progenitors could be defined via gene expression. As part of a broader screen to be described elsewhere, we employed CRISPR-mediated transgenesis (*Kimura et al., 2014*; *Ota et al., 2016*) to target expression constructs to genes that had been shown to be differentially expressed in support cells (*Jiang et al., 2014*; *Steiner et al., 2014*). This method inserts a construct with a minimal *hsp70l1* promoter driving fluorescent protein expression to a targeted break, so that expression is regulated by adjacent genomic elements. During this screen, we found that targeted insertion in three genes (*sfrp1a*, *tnfsf10l3*, and *sost*) resulted in distinct spatial expression in primary neuromast support cells: a transgene inserted in the *sfrp1a* locus is restricted to the most peripheral support cells (Peripheral cells; *Figure 2A*); insertion in *tnfsf10l3* is more broadly expressed throughout the periphery but is enriched in anteroposterior support cells (AP cells; *Figure 2C*); and insertion in *sost* is limited to the dorsal and ventral support cells (DV cells; *Figure 2E*). Secondary neuromasts are oriented orthogonally to primary neuromasts (Lopez-Schier et a., 2004); we found that the position of the distinct support cell populations are correspondingly rotated (*Figure 2—figure supplement 1*). We also generated GFP lines for each insertion site. We did not observe GFP labeling in hair cells in stable lines (*Figure 2—figure supplement 2*).

We used these distinct expression patterns to examine whether there were any functional differences between marked support cells with respect to their ability to serve as hair cell progenitors. To this end, we generated stable transgenic lines for all three loci: Tg[*sfrp1a*:nlsEos][w217]; Tg[*tnfsf10l3*:nlsEos][w218]; and Tg[*sost*:nlsEos][w215] (hereafter known as *sfrp1a*:nlsEos, *tnfsf10l3*:nlsEos, and *sost*:nlsEos, respectively). Eos is a photoconvertible protein that switches from green to red fluorescence (shown in magenta throughout this paper) after exposure to UV light (*Wiedenmann et al., 2004*). The converted protein is stable for months. Its nuclear localization presumably protects it from degradative elements in the cytoplasm, allowing for a more permanent label than a standard fluorescent reporter (*Cruz et al., 2015*; *McMenamin et al., 2014*). We could thus chase this label from support cell to hair cell if these cells serve as hair cell progenitors, as hair cells that derived from these support cells would have converted Eos in their nuclei. To ensure that expression of the nlsEos transgene didn't disrupt hair cell development or regeneration, we used the vital dye FM 1-43FX to label hair cells in transgenic fish and their non-transgenic siblings. In all three lines, we observed no significant difference between transgenic fish and their siblings (*Figure 2—figure supplement 3*). Because the uptake of FM 1-43FX requires proper mechanotransduction and can thus be used as a proxy for proper hair cell function (*Seiler and Nicolson, 1999*; *Gale et al., 2001*; *Meyers et al., 2003*), we conclude that expression of these nlsEos transgenes does not impact hair cell number or function during development or regeneration.

We first examined how these different support cell populations contributed to hair cell development and turnover under homeostatic conditions. All three nlsEos lines were crossed to a hair cell-specific transgenic line (Tg[Brn3c:GAP43-GFP][s356t] (*Xiao et al., 2005*), hereafter known as brn3c:GFP) in order to distinguish hair cell nuclei. Eos in support cells was photoconverted at 5 dpf and larvae were fixed and immunostained for GFP either immediately or at 8 dpf. At 5 dpf, 19% of hair cells were labeled with Eos expressed by the Peripheral cell transgene, and this number remained the same by 8 dpf (*Figure 2B*; p=0.7047). Eos from the AP cell transgene labeled about 6% of hair cells at both 5 and 8 dpf (*Figure 2D*; p=0.9668). Since there is no change over the three-day span, neither of these populations are responsible for generating new hair cells under homeostatic conditions. In contrast, the amount of hair cells labeled with photoconverted Eos from the DV cell transgene increased from 39% to 56% over that three-day span (*Figure 2F*; p<0.0001). Thus, the DV cell population seems to be predominantly involved in ongoing hair cell generation during homeostasis.

We next used these transgenic lines to determine whether there was any functional difference between these support cell subpopulations regarding their ability to serve as hair cell progenitors

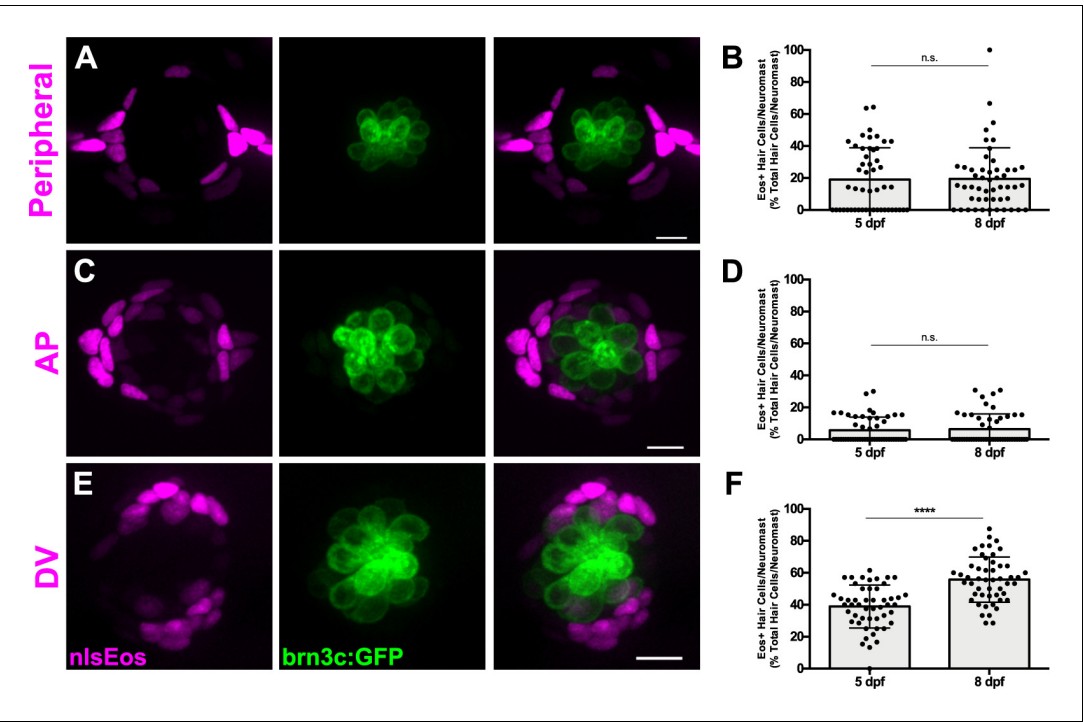

**Figure 2.** Genetic labeling of distinct support cell populations. (A, C, E) Maximum projections of neuromasts from *sfrp1a*:nlsEos (Peripheral, A), *tnfsf10l3*:nlsEos (AP, C), and *sost*:nlsEos (DV, E) fish. Converted nlsEos-positive cells are shown in magenta, and brn3c:GFP-positive hair cells are shown in green. Scale bar = 10 μm. (B, D, F) Percentage of hair cells per neuromast labeled by Peripheral (B), AP (D), and DV cells (F) at 5 and 8 dpf. (B) 5 dpf: 19.04 ± 19.86, n = 50 neuromasts (10 fish); 8 dpf: 19.46 ± 19.44, n = 50 neuromasts (10 fish); mean ± SD; Mann Whitney U test, p=0.7047. (D) 5 dpf: 5.71 ± 8.22, n = 50 neuromasts (10 fish); 8 dpf: 6.36 ± 9.57, n = 50 neuromasts (10 fish); mean ± SD; Mann Whitney U test, p=0.9668. (F) 5 dpf: 38.93 ± 13.46, n = 50 neuromasts (10 fish); 8 dpf: 55.78 ± 14.13, n = 50 neuromasts (10 fish); mean ± SD; Mann Whitney U test, p<0.0001.

DOI: https://doi.org/10.7554/eLife.43736.005

The following source data and figure supplements are available for figure 2:

**Source data 1.** Genetic labeling of distinct support cell populations
DOI: https://doi.org/10.7554/eLife.43736.010

**Figure supplement 1.** Asymmetry of support cell transgene expression in secondary neuromasts is orthogonal to primary neuromasts.
DOI: https://doi.org/10.7554/eLife.43736.006

**Figure supplement 2.** Support cell transgenes are not expressed in hair cells.
DOI: https://doi.org/10.7554/eLife.43736.007

**Figure supplement 3.** Expression of support cell transgenes does not alter hair cell development or regeneration.
DOI: https://doi.org/10.7554/eLife.43736.008

**Figure supplement 3—source data 1.** Expression of support cell transgenes does not alter hair cell development or regeneration
DOI: https://doi.org/10.7554/eLife.43736.009

during regeneration. Each of the nlsEos lines were once again crossed to brn3c:GFP fish in order to distinguish hair cell nuclei. Eos in support cells was photoconverted at 5 dpf, and larvae were subjected to neomycin-induced hair cell ablation and then fixed and immunostained for GFP at 72 hpt (*Figure 3A*). Only 4% of regenerated hair cells were derived from the Peripheral cell population, whereas the AP cell and DV cell populations contributed significantly more, generating 20% and 61% of regenerated hair cells, respectively (*Figure 3B–D*, arrowheads; *Figure 3E*; p=0.003 [Peripheral vs. AP], p<0.0001 [Peripheral/AP vs. DV]). In order to ensure that this difference in Eos incorporation was not simply due to relative proportion of available Eos-positive support cells, we counted the number of Eos-positive support cells in each transgenic line at 5 dpf, prior to hair cell ablation.

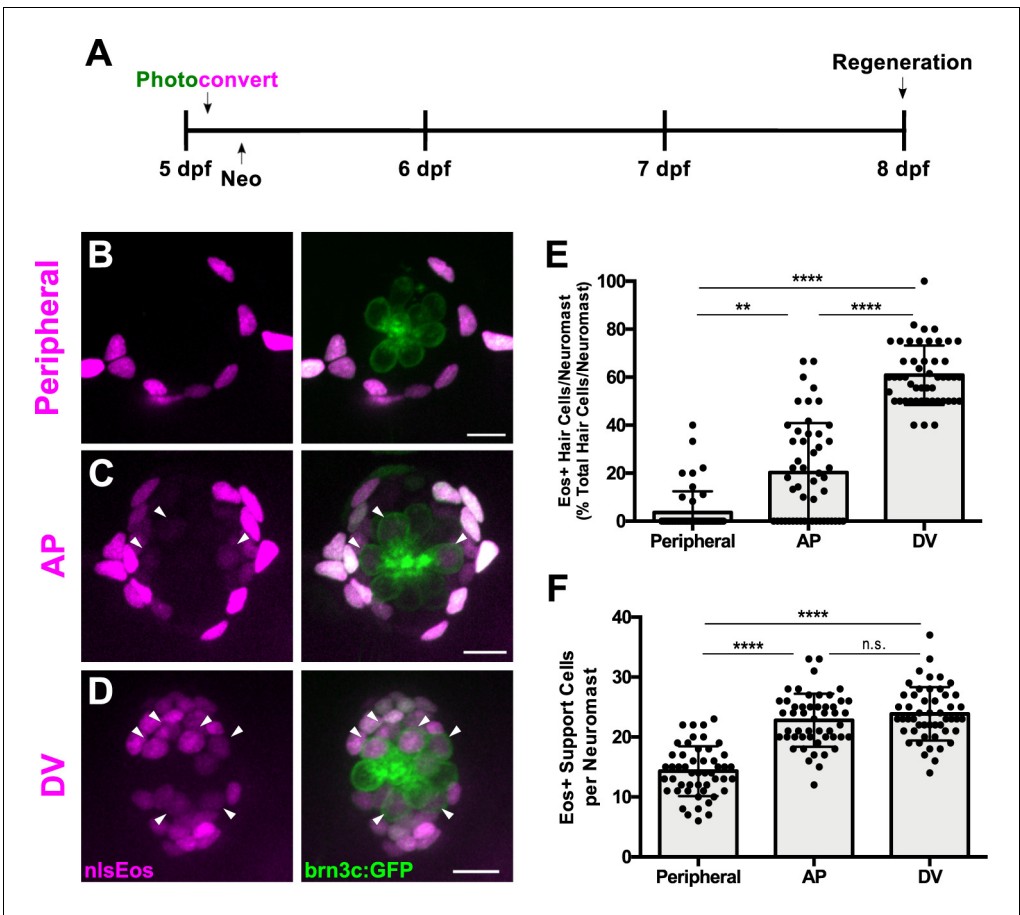

**Figure 3.** Distinct support cell populations have different regenerative capacities. (A) Timeline of nlsEos fate mapping experiment. Fish were photoconverted at 5 dpf, treated with neomycin, then fixed and imaged 72 hr post treatment (8 dpf). (B, C, D) Maximum projections of neuromasts from *sfrp1a*:nlsEos (Peripheral, B), *tnfsf10l3*: nlsEos (AP, C), and *sost*:nlsEos (DV, D) fish following photoconversion and hair cell regeneration. Converted nlsEos-positive cells are shown in magenta, and brn3c:GFP-positive hair cells are shown in green. Arrowheads indicate nlsEos-positive hair cells. Scale bar = 10 μm. (E) Percentage of hair cells per neuromast labeled by nlsEos following regeneration. *Sfrp1a*:nlsEos (Peripheral): 3.59 ± 8.87, n = 50 neuromasts (10 fish); *tnfsf10l3*:nlsEos (AP): 20.28 ± 20.58, n = 50 neuromasts (10 fish); *sost*:nlsEos (DV): 60.87 ± 12.37, n = 50 neuromasts (10 fish); mean ± SD; Kruskal-Wallis test, Dunn's post-test, p=0.003 (Peripheral vs. AP), p<0.0001 (Peripheral vs. DV, AP vs. DV). (F) Total nlsEos-positive support cells per neuromast prior to hair cell ablation. *Sfrp1a*:nlsEos (Peripheral): 14.30 ± 4.17, n = 50 neuromasts (10 fish); *tnfsf10l3*:nlsEos (AP): 22.8 ± 4.40, n = 50 neuromasts (10 fish); *sost*:nlsEos (DV): 23.86 ± 4.45, n = 50 neuromasts (10 fish); mean ± SD; Kruskal-Wallis test, Dunn's post-test, p<0.0001 (Peripheral vs. AP, Peripheral vs. DV), p>0.9999 (AP vs. DV).

DOI: https://doi.org/10.7554/eLife.43736.011

The following source data is available for figure 3:

**Source data 1.** Distinct support cell populations have different regenerative capacities
DOI: https://doi.org/10.7554/eLife.43736.012

There were about half as many Peripheral cells relative to the other two populations, but no significant difference between the number of AP cells and the number of DV cells (*Figure 3F*; Peripheral = 14.30 ± 4.17; AP = 22.8 ± 4.40; DV = 23.86 ± 4.45; p<0.0001 [Peripheral vs. AP/DV], p>0.9999 [AP vs. DV]). Thus, the difference in regenerative capacity between these populations is not simply a reflection of the number of available cells, but rather of differences in the progenitor identity of the populations.

## Inhibition of notch signaling differentially impacts support cell subpopulations

Notch-mediated lateral inhibition plays a crucial role in ensuring the proper number of hair cells are regenerated, and inhibition of Notch signaling following hair cell damage dramatically increases the number of regenerated hair cells (*Ma et al., 2008*; *Wibowo et al., 2011*; *Romero-Carvajal et al., 2015*). Thus, we examined how Notch inhibition impacted the progenitor function of our three support cell populations. We crossed all of our nlsEos lines to the brn3c:GFP line, and treated double-positive larvae with 50 µM LY411575 (LY), a potent γ-secretase inhibitor (*Mizutari et al., 2013*; *Romero-Carvajal et al., 2015*), for 24 hr immediately following neomycin treatment. Fish were fixed at 72 hpt and immunostained for GFP. In all three lines, after Notch inhibition (Neo/LY) there were roughly twice as many hair cells as control fish (Neo) (*Figure 4A–B,E–F,I–J*; p<0.0001 [all lines]), consistent with previous studies. The small number of Peripheral cell-derived hair cells was no different between LY-treated fish and non-treated fish (*Figure 4C*; 0.62 ± 1.28 [Neo] vs. 1.15 ± 2.16 [Neo/LY]; p=0.2481). By contrast, the number of nlsEos-positive hair cells from both AP and DV cells increased in fish treated with LY. Moreover, while the number of nlsEos-positive hair cells derived from DV cells doubled in LY-treated fish (*Figure 4K*; 7.40 ± 2.13 [Neo] vs. 15.25 ± 6.36 [Neo/LY]; p<0.0001), those

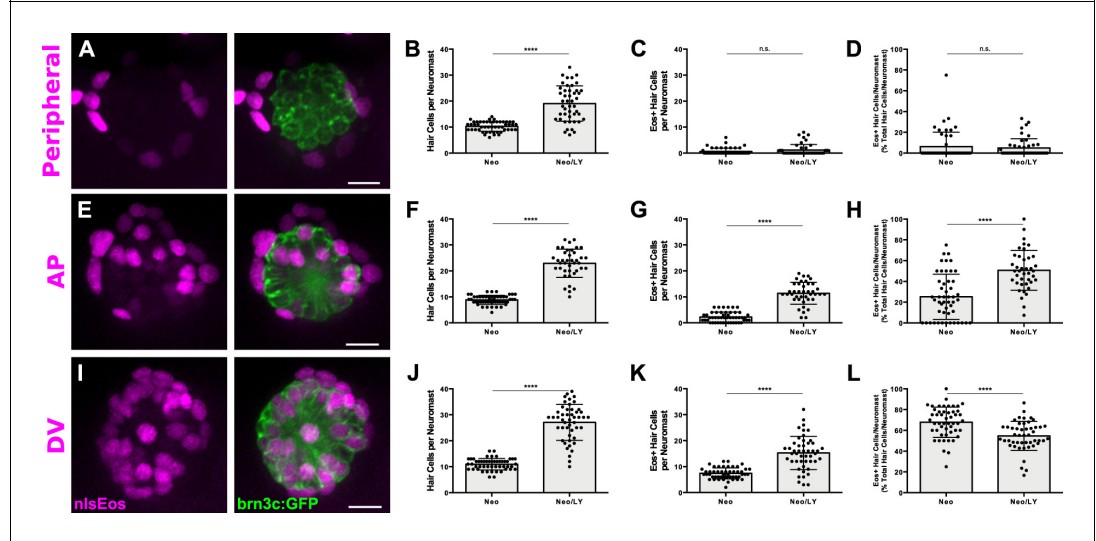

**Figure 4.** Notch signaling differentially regulates support cell populations. (A, E, I) Maximum projections of neuromasts expressing *sfrp1a*:nlsEos (Peripheral, A), *tnfsf10l3*:nlsEos (AP, E), and *sost*:nlsEos (DV, I) following Notch-inhibited hair cell regeneration. Converted nlsEos-positive cells are shown in magenta, and brn3c:GFP-positive hair cells are shown in green. Scale bar = 10 µm. (B) Total number of hair cells per neuromast in *sfrp1a*:nlsEos fish following hair cell regeneration. Neo: 10.28 ± 1.88, n = 50 neuromasts (10 fish); Neo/LY: 19.07 ± 6.79, n = 46 neuromasts (10 fish); mean ± SD; Mann Whitney U test, p<0.0001. (C) *Sfrp1a*:nlsEos-positive hair cells per neuromast following hair cell regeneration. Neo: 0.62 ± 1.28, n = 50 neuromasts (10 fish); Neo/LY: 1.15 ± 2.16, n = 46 neuromasts (10 fish); mean ± SD; Mann Whitney U test, p=0.2481. (D) Percentage of *sfrp1a*:nlsEos-labeled hair cells per neuromast following hair cell regeneration. Neo: 6.31 ± 13.83, n = 50 neuromasts (10 fish); Neo/LY: 4.95 ± 8.82, n = 46 neuromasts (10 fish); mean ± SD; Mann Whitney U test, p=0.5148. (F) Total number of hair cells per neuromast in *tnfsf10l3*:nlsEos fish following hair cell regeneration. Neo: 8.84 ± 1.75, n = 50 neuromasts (10 fish); Neo/LY: 22.93 ± 5.45, n = 40 neuromasts (eight fish); mean ± SD; Mann Whitney U test, p<0.0001. (G) *Tnfsf10l3*:nlsEos-positive hair cells per neuromast following hair cell regeneration. Neo: 2.22 ± 1.94, n = 50 neuromasts (10 fish); Neo/LY: 11.38 ± 4.23, n = 40 neuromasts (eight fish); mean ± SD; Mann Whitney U test, p<0.0001. (H) Percentage of *tnfsf10l3*:nlsEos-labeled hair cells per neuromast following hair cell regeneration. Neo: 25.19 ± 21.72, n = 50 neuromasts (10 fish); Neo/LY: 50.68 ± 19.23, n = 40 neuromasts (eight fish); mean ± SD; Mann Whitney U test, p<0.0001. (J) Total number of hair cells per neuromast in *sost*:nlsEos fish following hair cell regeneration. Neo: 10.94 ± 2.23, n = 50 neuromasts (10 fish); Neo/LY: 27.06 ± 6.90, n = 48 neuromasts (10 fish); mean ± SD; Mann Whitney U test, p<0.0001. (K) *Sost*:nlsEos-positive hair cells per neuromast following hair cell regeneration. Neo: 7.40 ± 2.13, n = 50 neuromasts (10 fish); Neo/LY: 15.25 ± 6.36, n = 48 neuromasts (10 fish); mean ± SD; Mann Whitney U test, p<0.0001. (L) Percentage of *sost*:nlsEos-labeled hair cells per neuromast following hair cell regeneration. Neo: 67.86 ± 14.63, n = 50 neuromasts (10 fish); Neo/LY: 54.69 ± 14.01, n = 48 neuromasts (10 fish); mean ± SD; Mann Whitney U test, p<0.0001.

DOI: https://doi.org/10.7554/eLife.43736.013

The following source data is available for figure 4:

**Source data 1.** Notch signaling differentially regulates support cell populations
DOI: https://doi.org/10.7554/eLife.43736.014

derived from AP cells increased roughly five-fold (*Figure 4G*; 2.22 ± 1.94 [Neo] vs. 11.38 ± 4.23 [Neo/LY]; p<0.0001). As a consequence, the percentage of hair cells derived from DV cells decreased correspondingly (*Figure 4L*; 67.86 ± 14.63 [Neo] vs. 54.69 ± 14.01 [Neo/LY]; p<0.0001), whereas those derived from AP cells doubled (*Figure 4H*; 25.19 ± 21.72 [Neo] vs. 50.68 ± 19.23 [Neo/LY]; p<0.0001). These data suggest that generation of hair cells from both the AP and DV populations is regulated by Notch signaling (with the AP population being regulated to a greater extent), whereas Peripheral cells are not responsive to Notch signaling.

## Selective ablation of DV cells reduces hair cell regeneration

Since the DV cell population generates roughly 60% of regenerated hair cells, we sought to determine whether these cells were required for hair cell regeneration. To this end, we generated a transgenic line in which an enhanced-potency nitroreductase (epNTR; *Tabor et al., 2014*) fused to GFP was introduced into the *sost* locus using CRISPR (Tg[*sost*:epNTR-GFP]$^{w216}$, hereafter known as *sost*:NTR-GFP). Nitroreductase is a bacterial enzyme that selectively binds its prodrug Metronidazole (Mtz), converting Mtz into toxic metabolites that kill the cells expressing it (*Curado et al., 2007*). We then compared the extent of *sost:NTR-GFP* expression in DV cells, as defined by the *sost:nlsEos* transgene. At three dpf, soon after the initiation of transgene expression, we see considerable overlap between NTR-GFP and nlsEos. All NTR-GFP +cells were also positive for nlsEos, while an additional subset of cells expressed nlsEos alone. When we compared expression at five dpf, the size of the double-positive (NTR-GFP+; nlsEos+) population did not change, whereas the number of cells expressing nlsEos alone increased significantly, occupying a more central location (*Figure 5A–B*, arrowheads; *Figure 5C*; NTR-GFP/nlsEos: 9.04 ± 2.39 [3 dpf] vs. 8.47 ± 2.27 [5 dpf]; nlsEos only: 6.10 ± 2.27 [3 dpf] vs. 10.86 ± 2.72 [5 dpf]; p>0.9999 [NTR-GFP/nlsEos], p<0.0001 [nlsEos only]). These observations are consistent with the idea that both transgenes initiate expression at the same time, but that nlsEos protein is retained longer than NTR-GFP protein as cells mature and as a result, NTR-GFP is expressed in a subset of DV cells. We next tested to the efficacy of DV cell ablation at 3 and 5 dpf. Treatment of these fish with 10 mM Mtz for 8 hr was sufficient to ablate the majority of NTR-GFP cells. Treating fish with Mtz for 8 hr at five dpf (Mtz5) slightly but significantly decreased the number of support cells solely expressing nlsEos by about 13%. Treating fish with Mtz for 8 hr at three dpf, followed by a second 8 hr Mtz treatment at five dpf (Mtz3/5) decreased the number of solely nlsEos-positive cells even further, by about 40% (*Figure 5D–G*; Mock: 11.18 ± 2.04; Mtz5: 9.72 ± 2.03; Mtz3/5: 6.76 ± 2.12; p=0.0288 [Mock vs. Mtz5], p<0.0001 [Mock vs. Mtz3/5, Mtz5 vs. Mtz3/5]).

We next tested the impact of DV cell ablation on hair cell regeneration. We compared two groups: neomycin exposure followed by Mtz treatment at 5 dpf (Neo/Mtz5), compared to Mtz treatment at 3 dpf, then neomycin treatment at 5 dpf followed by a second Mtz treatment (Mtz3/Neo/Mtz5; *Figure 6A*). For both groups, nlsEos was photoconverted at 5 dpf, just prior to neomycin treatment, and larvae were fixed at 72 hpt and immunostained for GFP and Parvalbumin (to label NTR-GFP+ cells and hair cells, respectively). The Neo/Mtz5 treatment resulted in a small but significant reduction in both hair cells and nlsEos-positive hair cells per neuromast relative to normal regeneration (*Figure 6B–C,E–F*; Total hair cells: 11.73 ± 2.10 [Neo] vs. 9.33 ± 1.88 [Neo/Mtz5]; p=0.0001; nlsEos+ hair cells: 7.78 ± 2.36 [Neo] vs. 4.90 ± 2.02 [Neo/Mtz5]; p=0.0003). The Mtz3/Neo/Mtz5-treated larvae exhibited even fewer hair cells per neuromast (*Figure 6D,E*; 11.73 ± 2.10 [Neo] vs. 7.52 ± 1.74 [Mtz3/Neo/Mtz5]; p<0.0001), with nlsEos-labeled hair cells decreased to a mere 14% of total regenerated hair cells (*Figure 6G*; 65.81 ± 14.89 [Neo] vs. 14.29 ± 18.10 [Mtz3/Neo/Mtz5]; p<0.0001). Importantly, Mtz treatment of siblings without the *sost*:NTR-GFP transgene had no impact on hair cell regeneration (*Figure 6—figure supplement 1*; Neo: 9.5 ± 1.50; Mtz3/Neo/Mtz5: 9.98 ± 1.51; p=0.2317). Transgenes also had no obvious effect on hair cell development or regeneration in the absence of Mtz either alone or in combination (*Figure 6—figure supplement 2*). Thus, ablation of DV cells reduces the number of hair cells regenerated.

We then examined how Notch signaling impacted hair cell regeneration in the context of DV cell ablation. We treated *sost*:NTR-GFP larvae with 50 µM LY for 24 hr following ablation (Mtz3/Neo/Mtz5/LY) and assayed hair cell number at 72 hpt (*Figure 7A*). As expected, the number of regenerated hair cells increased significantly after LY treatment in all groups (*Figure 7B–F*; p<0.0001), and DV cell ablation significantly decreased hair cell regeneration (*Figure 7B,D,F*; 9.42 ± 1.85 [Neo] vs. 6.86 ± 1.76 [Mtz3/Neo/Mtz5]; p=0.0058). However, LY treatment following Mtz ablation resulted in

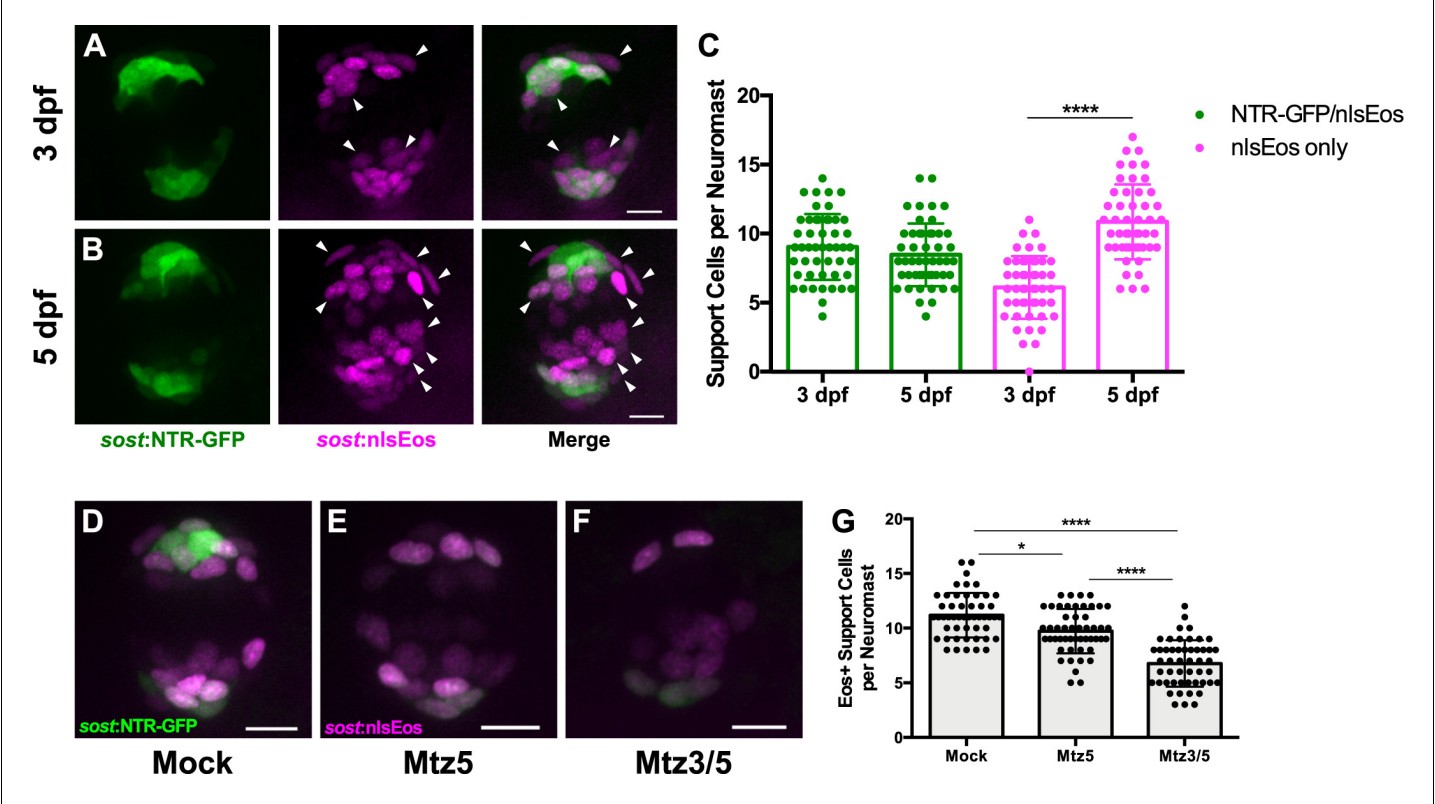

**Figure 5.** Differences in overlap between *sost*:NTR-GFP and *sost*:nlsEos populations. (A–B) Maximum projections of neuromasts from *sost*:NTR-GFP; *sost*:nlsEos fish at 3 dpf (A) and 5 dpf (B). *Sost*:NTR-GFP cells are shown in green and *sost*:nlsEos cells are shown in magenta. Arrowheads indicate cells expressing *sost*:nlsEos but not *sost*:NTR-GFP. Scale bar = 10 μm. (C) Support cells per neuromast expressing either NTR-GFP and nlsEos (green) or nlsEos only (magenta) at 3 dpf and 5 dpf. NTR-GFP/nlsEos: 9.04 ± 2.39 (3 dpf) vs. 8.47 ± 2.27 (5 dpf), n = 49 neuromasts each (10 fish each); nlsEos only: 6.10 ± 2.27 (3 dpf) vs. 10.86 ± 2.72 (5 dpf), n = 49 neuromasts each (10 fish each); mean ± SD; Kruskal-Wallis test, Dunn's post-test, p>0.9999 (NTR-GFP/nlsEos 3 dpf vs. 5 dpf), p<0.0001 (nlsEos only 3 dpf vs. 5 dpf). (D–F) Maximum projections of neuromasts from *sost*:NTR-GFP; *sost*:nlsEos fish following mock treatment (D; Mock), Mtz at 5 dpf (E; Mtz5), and Mtz at 3 dpf and 5 dpf (F; Mtz3/5). *Sost*:NTR-GFP cells are shown in green and *sost*:nlsEos cells are shown in magenta. Scale bar = 10 μm. (G) Support cells per neuromast solely expressing *sost*:nlsEos following Mtz treatment. Mock: 11.18 ± 2.04, n = 50 neuromasts (10 fish); Mtz5: 9.72 ± 2.03, n = 50 neuromasts (10 fish); Mtz3/5: 6.76 ± 2.12, n = 50 neuromasts (10 fish); mean ±SD; Kruskal-Wallis test, Dunn's post-test, p=0.0288 (Mock vs. Mtz5), p<0.0001 (Mock vs. Mtz3/5, Mtz5 vs. Mtz3/5).
DOI: https://doi.org/10.7554/eLife.43736.015

The following source data is available for figure 5:

**Source data 1.** Differences in overlap between *sost*:NTR-GFP and *sost*:nlsEos populations
DOI: https://doi.org/10.7554/eLife.43736.016

significantly fewer regenerated hair cells than LY alone (*Figure 7C,E,F*; 21.08 ± 4.42 [Neo/LY] vs. 15.06 ± 3.51 [Mtz3/Neo/Mtz5/LY]; p=0.0029), indicating that inhibiting Notch signaling cannot fully compensate for the loss of the DV population.

## AP and DV cells define separate progenitor populations

While the DV population generates roughly 60% of hair cells after damage, the other 40% must derive from a different population. Consistent with this observation, reduction of DV cells by Mtz treatment only partially blocks new hair cell formation, indicating that there must be additional progenitor populations. We believed that AP cells could define this additional population, since they were capable of generating roughly 20% of regenerated hair cells (*Figure 3E*). However, there may be some overlap between the expression of *tnfsf10l3*:nlsEos defining the AP domain and *sost*:nlsEos defining the DV domain. When we crossed the *tnfsf10l3*:nlsEos and *sost*:nlsEos lines together, we found that roughly 88% of regenerated hair cells were nlsEos positive when larvae expressed both transgenes, compared to 65% from *sost*:nlsEos alone and 28% from *tnfsf10l3*:nlsEos alone

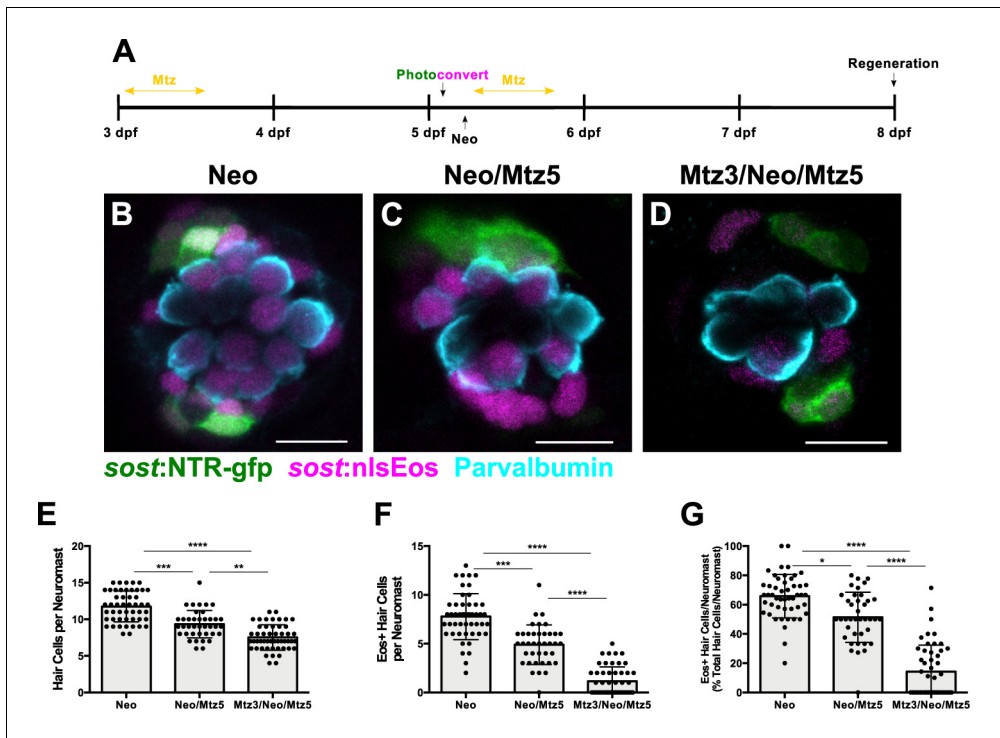

**Figure 6.** Ablation of DV cells decreases number of regenerated hair cells. (A) Timeline of DV cell-ablation experiment. Larvae were treated with Mtz at 3 dpf, photoconverted, then treated with neomycin, then treated with Mtz again at 5 dpf, and fixed and immunostained at 72 hpt (8 dpf). (B–D) Maximum projections of neuromasts from *sost*:NTR-GFP; *sost*:nlsEos fish following neomycin (B; Neo), neomycin and Mtz (C; Neo/Mtz5), and Mtz, neomycin, and Mtz treatments (D; Mtz3/Neo/Mtz5). *Sost*:NTR-GFP cells are shown in green, *sost*:nlsEos cells are shown in magenta, and anti-Parvalbumin-stained hair cells are shown in cyan. Scale bar = 10 μm. (E) Total hair cells per neuromast following regeneration. Neo: 11.73 ± 2.10, n = 49 neuromasts (10 fish); Neo/Mtz5: 9.33 ± 1.88, n = 39 neuromasts (8 fish); Mtz3/Neo/Mtz5: 7.52 ± 1.74, n = 50 neuromasts (10 fish); mean ± SD; Kruskal-Wallis test, Dunn's post-test, p=0.0001 (Neo vs. Neo/Mtz5), p<0.0001 (Neo vs. Mtz3/Neo/Mtz5), p=0.0016 (Neo/Mtz5 vs. Mtz3/Neo/Mtz5). (F) *Sost*:nlsEos-positive hair cells per neuromast following regeneration. Neo: 7.78 ± 2.36, n = 49 neuromasts (10 fish); Neo/Mtz5: 4.90 ± 2.02, n = 39 neuromasts (eight fish); Mtz3/Neo/Mtz5: 1.16 ± 1.46, n = 50 neuromasts (10 fish); mean ± SD; Kruskal-Wallis test, Dunn's post-test, p=0.0003 (Neo vs. Neo/Mtz5), p<0.0001 (Neo vs. Mtz3/Neo/Mtz5, Neo/Mtz5 vs. Mtz3/Neo/Mtz5). (G) Percentage of hair cells per neuromast labeled by *sost*:nlsEos following regeneration. Neo: 65.81 ± 14.89, n = 49 neuromasts (10 fish); Neo/Mtz5: 51.40 ± 17.17, n = 39 neuromasts (8 fish); Mtz3/Neo/Mtz5: 14.29 ± 18.10, n = 50 neuromasts (10 fish); mean ± SD; Kruskal-Wallis test, Dunn's post-test, p=0.0147 (Neo vs. Neo/Mtz5), p<0.0001 (Neo vs. Mtz3/Neo/Mtz5, Neo/Mtz5 vs. Mtz3/Neo/Mtz5).

DOI: https://doi.org/10.7554/eLife.43736.017

The following source data and figure supplements are available for figure 6:

**Source data 1.** Ablation of DV cells decreases number of regenerated hair cells
DOI: https://doi.org/10.7554/eLife.43736.022

**Figure supplement 1.** Mtz treatment does not inherently impact hair cell regeneration.
DOI: https://doi.org/10.7554/eLife.43736.018

**Figure supplement 1—source data 1.** Mtz treatment does not inherently impact hair cell regeneration
DOI: https://doi.org/10.7554/eLife.43736.019

**Figure supplement 2.** Hair cell development and regeneration is unaffected in fish expressing both *sost*:NTR-GFP and *sost*:nlsEos.
DOI: https://doi.org/10.7554/eLife.43736.020

**Figure supplement 2—source data 1.** Hair cell development and regeneration is unaffected in fish expressing both*sost*:NTR-GFP and *sost*:nlsEos
DOI: https://doi.org/10.7554/eLife.43736.021

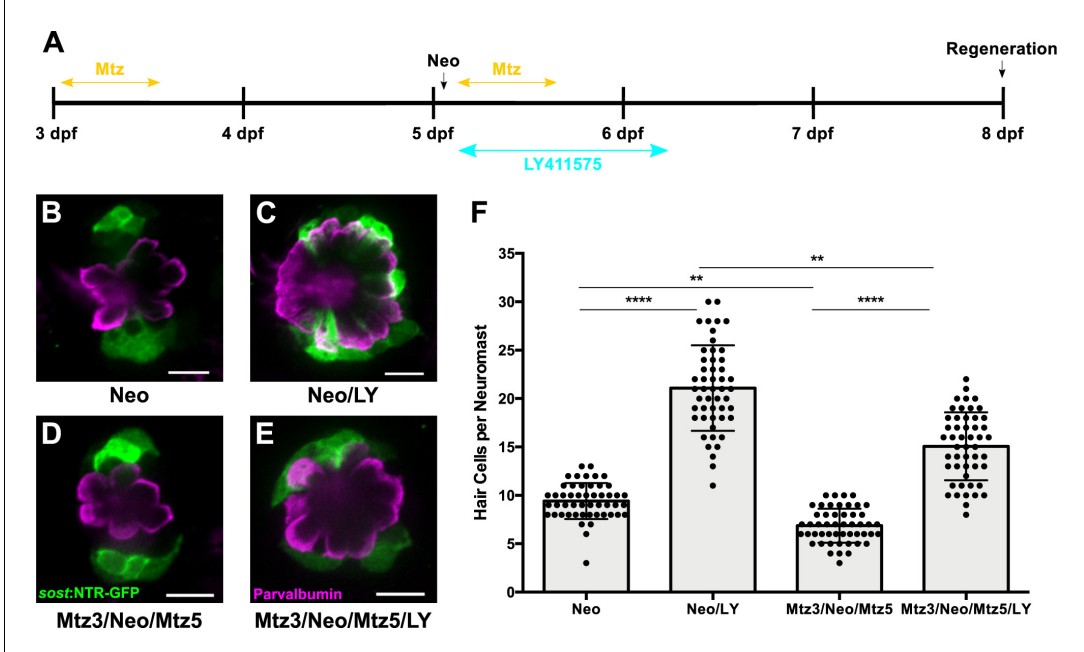

**Figure 7.** DV cell-ablation reduces the number of supernumerary hair cells formed during Notch-inhibited hair cell regeneration. (A) Timeline of dual DV cell-ablation, Notch-inhibition experiment. *Sost*:NTR-GFP larvae were treated with Mtz at 3 dpf, treated with neomycin at 5dpf, then co-treated with Mtz and LY411575 for 8 hr, then washed out and treated with LY411575 for 16 additional hours (24 hr total LY). (B–E) Maximum projections of *sost*:NTR-GFP neuromasts following normal hair cell regeneration (B; Neo), Notch-inhibited hair cell regeneration (C; Neo/LY), DV cell-ablated hair cell regeneration (D; Mtz3/Neo/Mtz5), and DV cell-ablated and Notch-inhibited hair cell regeneration (E; Mtz3/Neo/Mtz5/LY). *Sost*:NTR-GFP cells are shown in green, and anti-Parvalbumin immunostained hair cells are shown in magenta. Scale bar = 10 μm. (F) Total number of hair cells per neuromast following hair cell regeneration. Neo: 9.42 ± 1.85, n = 50 neuromasts (10 fish); Neo/LY: 21.08 ± 4.42, n = 50 neuromasts (10 fish); Mtz3/Neo/Mtz5: 6.86 ± 1.76, n = 50 neuromasts (10 fish); Mtz3/Neo/Mtz5/LY: 15.06 ± 3.51, n = 50 neuromasts (10 fish); mean ± SD; Kruskal-Wallis test, Dunn's post-test, p<0.0001 (Neo vs. Neo/LY; Mtz3/Neo/Mtz5 vs. Mtz3/Neo/Mtz5/LY), p=0.0058 (Neo vs. Mtz3/Neo/Mtz5), p=0.0029 (Neo/LY vs. Mtz3/Neo/Mtz5/LY).
DOI: https://doi.org/10.7554/eLife.43736.023

The following source data is available for figure 7:

**Source data 1.** DV cell-ablation reduces the number of supernumerary hair cells formed during Notch-inhibited hair cell regeneration
DOI: https://doi.org/10.7554/eLife.43736.024

(*Figure 8A–E*; p<0.0001). Thus, while not completely additive, these data suggest that the AP population is distinct from the DV population in terms of its progenitor function.

We next examined how the AP population would respond to the ablation of the DV population. We crossed the *tnfsf10l3*:nlsEos line to the *sost*:NTR-GFP line, sorted out double-positive larvae, and compared normal regeneration to that after Mtz treatment (Mtz3/Neo/Mtz5, since this had served to be the best treatment paradigm). As above, nlsEos was photoconverted at 5 dpf, immediately prior to neomycin treatment, and larvae were fixed at 72 hpt and immunostained for GFP and Parvalbumin. Mtz-ablated larvae had significantly fewer hair cells than non-ablated larvae, as in previous experiments (*Figure 9C*; 10.36 ± 1.60 [Neo] vs. 7.98 ± 1.74 [Mtz3/Neo/Mtz5]; p<0.0001), but the number of nlsEos-positive hair cells was no different between the two groups (*Figure 9A–B*, arrowheads; *Figure 9D*; 2.88 ± 1.83 [Neo] vs. 3.14 ± 1.43 [Mtz3/Neo/Mtz5]; p=0.3855). Thus, the AP population's progenitor function remains unchanged following DV ablation, providing further support that it is a separate progenitor population from the DV population.

## The DV population regenerates from other support cell subpopulations

When examining hair cell regeneration following DV cell ablation, we consistently noticed that there was an increase in NTR-GFP +cells at 72 hpt. This led us to hypothesize that DV cells were capable of regeneration even in the absence of hair cell damage. To test this, we first administered a 48 hr pulse of EdU (changing into fresh EdU solution after the first 24 hr) immediately following Mtz ablation at 5 dpf and fixed immediately after EdU washout. At 48 hr post ablation, we observed slightly

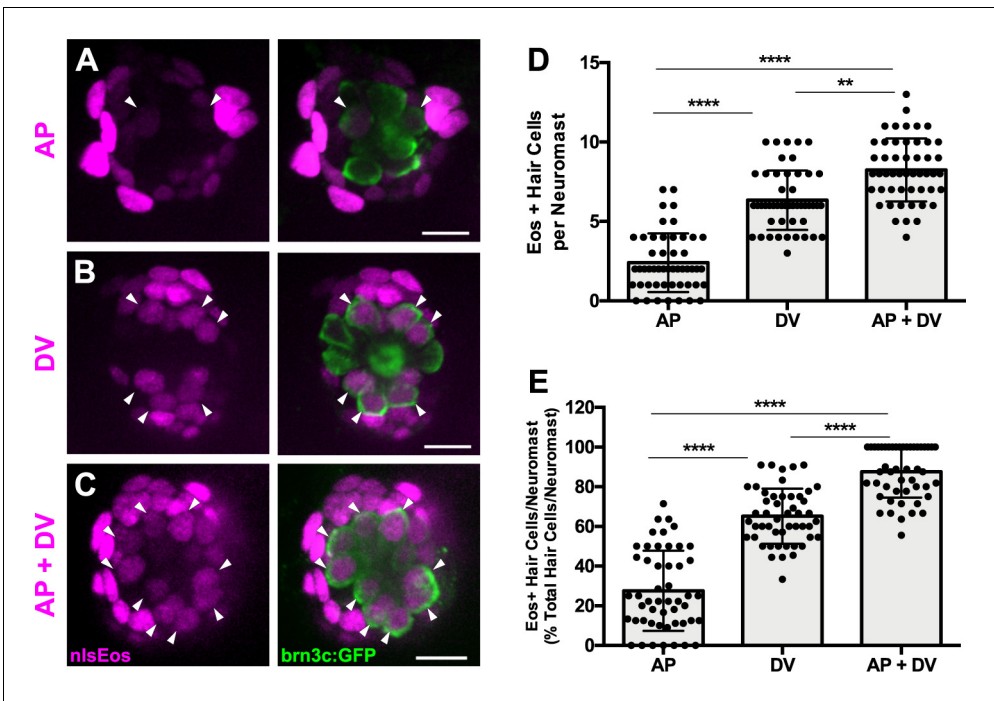

**Figure 8.** AP cells and DV cells define separate progenitor populations. (**A–C**) Maximum projections of neuromasts from *tnfsf10l3*:nlsEos (AP, (**A**), *sost*:nlsEos (DV, (**B**), and *tnfsf10l3*:nlsEos/*sost*:nlsEos fish (AP + DV, (**C**) following photoconversion and regeneration. Converted nlsEos-positive cells are shown in magenta, and brn3c: GFP-positive hair cells are shown in green. Arrowheads indicate nlsEos-positive hair cells. Scale bar = 10 µm. (**D**) Number of nlsEos-positive hair cells per neuromast in each of the nlsEos lines following regeneration. *Tnfsf10l3*: nlsEos (AP): 2.4 ± 1.84, n = 50 neuromasts (10 fish); *sost*:nlsEos (DV): 6.34 ± 1.87, n = 50 neuromasts (10 fish); *tnfsf10l3*:nlsEos/*sost*:nlsEos (AP + DV): 8.24 ± 1.99, n = 50 neuromasts (10 fish); mean ±SD; Kruskal-Wallis test, Dunn's post-test, p<0.0001 (AP vs. DV, AP vs. AP + DV), p=0.0031 (DV vs. AP + DV). (**E**) Percentage of hair cells per neuromast labeled by nlsEos lines following regeneration. AP: 27.59 ± 20.21, n = 50 neuromasts (10 fish); DV: 65.16 ± 13.89, n = 50 neuromasts (10 fish); AP + DV: 87.57 ± 13.02, n = 50 neuromasts (10 fish); mean ± SD; Kruskal-Wallis test, Dunn's post-test, p<0.0001 (all comparisons).

DOI: https://doi.org/10.7554/eLife.43736.025

The following source data is available for figure 8:

**Source data 1.** AP cells and DV cells define separate progenitor populations

DOI: https://doi.org/10.7554/eLife.43736.026

more than half the number of the NTR-GFP +cells relative to unablated larvae (*Figure 10C*; 8.94 ± 1.62 [Mock] vs. 5.34 ± 2.14 [Mtz]; p<0.0001). However, 58% of NTR-GFP +cells were EdU-positive in fish treated with Mtz, compared to just 15% in unablated larvae (*Figure 10A–B*, arrowheads; *Figure 10D*; p<0.0001). These results indicate that new DV cells arise from proliferation.

To determine the source of new DV cells, we crossed *sost*:NTR-GFP fish to our three different nlsEos lines. Double-transgenic fish were photoconverted at 5 dpf, Mtz-ablated, and then fixed at 72 hpt and immunostained for GFP. Following ablation, 56% of NTR-GFP+ cells expressed photoconverted nlsEos when DV cells were labeled, compared to 97% in unablated controls (*Figure 11A–B*, arrowheads; *Figure 11C*; p<0.0001). 31% of NTR-GFP+ cells expressed photoconverted nlsEos when Peripheral cells were labeled, compared to 6% in controls (*Figure 11D–E*, arrowheads; *Figure 11F*; p<0.0001) and 21% of NTR-GFP+ cells expressed photoconverted nlsEos when AP cells were labeled, compared to 7% in controls (*Figure 11G–H*, arrowheads; *Figure 11I*; p=0.0004). Thus, DV cells are capable of being replenished after Mtz ablation by other DV cells as well as by both AP and Peripheral cells.

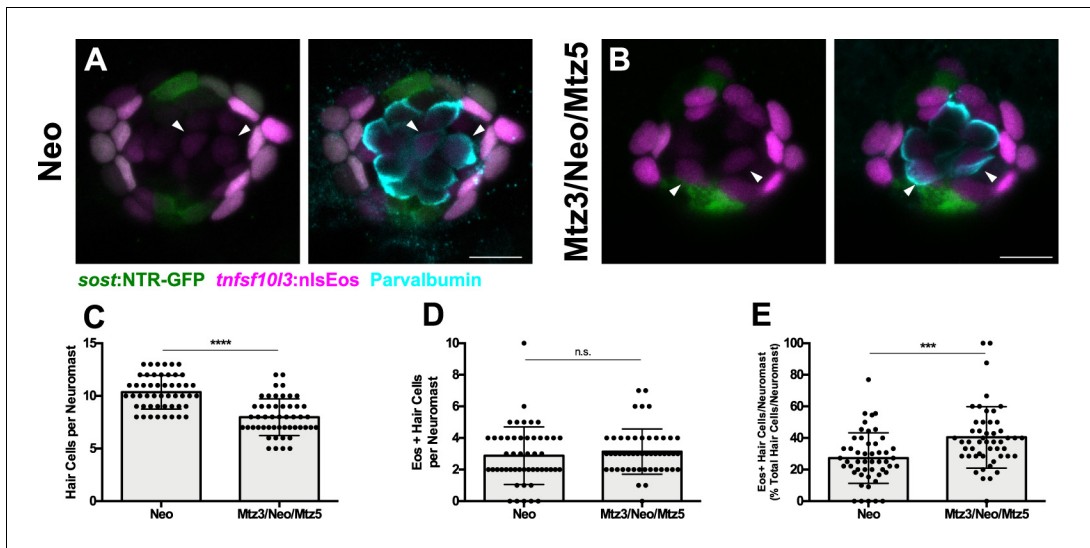

**Figure 9.** AP population doesn't compensate for the loss of the DV population during hair cell regeneration. (A–B) Maximum projections of *tnfsf10l3*:nlsEos; *sost*:NTR-GFP neuromasts following normal hair cell regeneration (A; Neo) or DV cell-ablated hair cell regeneration (B; Mtz3/Neo/Mtz5). *Sost*:NTR-GFP cells are shown in green, *tnfsf10l3*:nlsEos cells are shown in magenta, and anti-Parvalbumin-stained hair cells are shown in cyan. Arrowheads indicate nlsEos-positive hair cells. Scale bar = 10 μm. (C) Total number of hair cells per neuromast following hair cell regeneration. Neo: 10.36 ± 1.60, n = 50 neuromasts (10 fish); Mtz3/Neo/Mtz5: 7.98 ± 1.74, n = 50 neuromasts (10 fish); mean ± SD; Mann Whitney U test, p<0.0001. (D) Number of nlsEos-positive hair cells per neuromast following hair cell regeneration. Neo: 2.88 ± 1.83, n = 50 neuromasts (10 fish); Mtz3/Neo/Mtz5: 3.14 ± 1.43, n = 50 neuromasts (10 fish); mean ± SD; Mann Whitney U test, p=0.3855. (E) Percentage of hair cells per neuromast labeled by nlsEos following hair cell regeneration. Neo: 27.26 ± 16.00, n = 50 neuromasts (10 fish); Mtz3/Neo/Mtz5: 40.43 ± 19.44, n = 50 neuromasts (10 fish); mean ± SD; Mann Whitney U test, p=0.0002.

DOI: https://doi.org/10.7554/eLife.43736.027

The following source data is available for figure 9:

**Source data 1.** AP population doesn't compensate for the loss of the DV population during hair cell regeneration
DOI: https://doi.org/10.7554/eLife.43736.028

## Discussion

### Differences in hair cell progenitor identity among support cell populations

The data shown above indicate that there are at least three functionally distinct progenitor populations within the neuromast: (1) a highly regenerative, dorsoventral (DV) population, marked by *sost*:nlsEos, which generates the majority of regenerated hair cells; (2) an anteroposterior (AP) population, marked by *tnfsf10l3*:nlsEos, which also contributes to hair cell regeneration albeit to a far lesser extent than *sost*; and (3) a peripheral population (Peripheral), marked by *sfrp1a*:nlsEos, that does not contain hair cell progenitors (*Figure 12*). This model of high regenerative capacity in the dorsoventral region and low regenerative capacity in the anteroposterior region is consistent with the label-retaining studies performed by *Cruz et al. (2015)* as well as the BrdU-localization studies of *Romero-Carvajal et al. (2015)*. However, an examination of the overlap in expression between *sost*:nlsEos and *sost*:NTR-GFP reveals distinctions even amongst this DV progenitor population. We hypothesize that cells that express only nlsEos have matured from those that express both NTR-GFP and nlsEos. We posit that these more mature nlsEos cells serve as hair cell progenitors. Consistent with this idea, Mtz treatment at five dpf that spares nlsEos cells not expressing NTR-GFP has only a small effect on hair cell regeneration while Mtz treatment at both 3 and 5 dpf results in substantial reduction in hair cell regeneration.

While we have identified distinct hair cell progenitor populations (DV and AP), these populations do not account for all of the hair cell progenitors in the neuromast. The combination of these cells

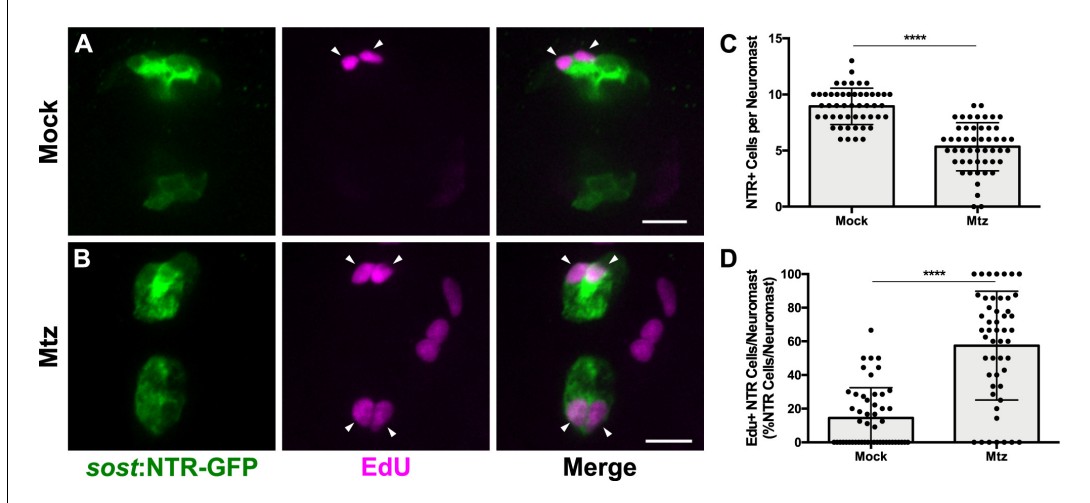

**Figure 10.** DV population regenerates via proliferation. (A–B) Maximum projections of neuromasts from *sost*:NTR-GFP fish either untreated (A; Mock) or treated with 10 mM Mtz (B; Mtz). *Sost*:NTR-GFP cells are shown in green and EdU-positive cells are shown in magenta. Arrowheads indicate EdU-positive *sost*:NTR-GFP cells. Scale bar = 10 µm. (C) Total number of *sost*:NTR-GFP cells per neuromast following DV cell regeneration. Mock: 8.94 ± 1.62, n = 50 neuromasts (10 fish); Mtz: 5.34 ± 2.14, n = 50 neuromasts (10 fish); mean ± SD; Mann Whitney U test, p<0.0001. (D) Percentage of *sost*:NTR-GFP cells per neuromast labeled by EdU following DV cell regeneration. Mock: 14.47 ± 17.95, n = 50 neuromasts (10 fish); Mtz: 57.49 ± 32.34, n = 50 neuromasts (10 fish); mean ± SD; Mann Whitney U test, p<0.0001.

DOI: https://doi.org/10.7554/eLife.43736.029

The following source data is available for figure 10:

**Source data 1.** DV population regenerates via proliferation
DOI: https://doi.org/10.7554/eLife.43736.030

generated 88% of new hair cells, meaning that the remaining 12% were derived from other sources. Furthermore, the AP population only accounted for 40% of new hair cells generated after neomycin treatment following DV cell ablation. Thus, there must be some other population (or populations) of support cells that are serving as hair cell progenitors that we have not labeled with our transgenic techniques. The best candidates for this role are centrally located support cells found ventral to hair cells, although the identity of these cells remains to be determined.

It is worth noting that Peripheral and DV lines (both nlsEos and NTR-GFP) were generated by targeted integration of the transgene into exons (see Materials and methods for details). Although CRISPR-mediated exonic integration has been shown to generate loss-of-function mutations (*Ota et al., 2016*), proper hair cell development and regeneration were unaffected in lines heterozygous for transgene insertion (*Figure 2—figure supplement 3*). Zebrafish expressing both nlsEos and NTR-GFP in DV cells (*Figures 5*, *6* and *11A–C*) would theoretically be null for *sost* function, yet these double positive larvae have the same number of hair cells during development (five dpf) and after hair cell regeneration as their non-transgenic and heterozygotic siblings (*Figure 6—figure supplement 2*). This would suggest that *sost* function is dispensable for hair cell development and regeneration, in spite of the contribution DV cells make to both processes. However, we did not formally test whether *sost* function was actually disrupted by transgene insertion, so it is possible that these double-positive larvae are not indicative of true *sost* loss-of-function or that there are mechanisms to compensate for the loss of *sost*.

It is possible that our transgenic lines do not reflect the corresponding gene expression of the locus of insertion, although this CRISPR-mediated transgenesis method has been shown to do so for other genes (*Kimura et al., 2014*; *Ota et al., 2016*). While we have not verified that the expression patterns of our transgenes correspond to those of the endogenous loci by in situ hybridization, a recent comprehensive analysis of gene expression in neuromast support cells (*Lush et al., 2019*) confirms that the endogenous transcripts of *sfrp1a*, *sost*, and *tnfsf10l3* have similar patterns to those of the transgenic insertions reported here. We stress that the purpose of this study is not to correlate progenitor function to specific gene function, but to examine the functional differences between

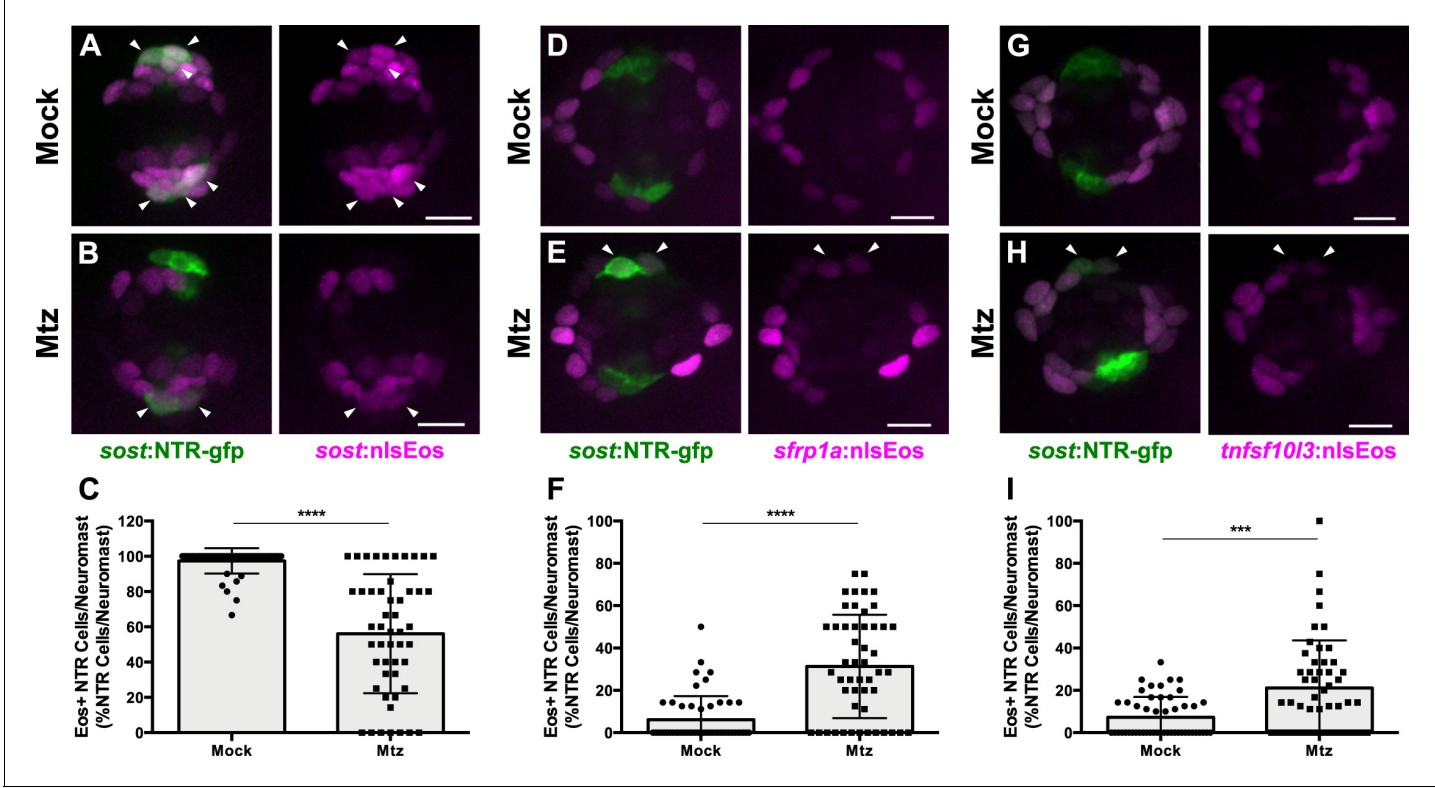

**Figure 11.** DV cells are replenished by other support cell populations. (**A–B, D–E, G–H**) Maximum projections of neuromasts expressing *sost*:NTR-GFP and *sost*:nlsEos (A–B), *sfrp1a*:nlsEos (D–E), and *tnfsf10l3*:nlsEos (G–H) in the absence of (A, D, G; Mock) or following Mtz-induced DV cell ablation (B, E, H; Mtz). *Sost*:NTR-GFP cells are shown in green and nlsEos-positive cells are shown in magenta. Arrowheads indicate nlsEos-positive *sost*:NTR-GFP cells. Scale bar = 10 µm. (**C**) Percentage of *sost*:NTR-GFP cells per neuromast labeled by *sost*:nlsEos following DV cell regeneration. Mock: 97.39 ± 7.14, n = 50 neuromasts (10 fish); Mtz: 56.09 ± 33.72, n = 50 neuromasts (10 fish); mean ± SD; Mann Whitney U test, p<0.0001. (**F**) Percentage of *sost*:NTR-GFP cells per neuromast labeled by *sfrp1a*:nlsEos following DV cell regeneration. Mock: 6.15 ± 11.14, n = 50 neuromasts (10 fish); Mtz: 31.27 ± 24.41, n = 50 neuromasts (10 fish); mean ± SD; Mann Whitney U test, p<0.0001. (**I**) Percentage of *sost*:NTR-GFP cells per neuromast labeled by *tnfsf10l3*:nlsEos following DV cell regeneration. Mock: 7.31 ± 9.55, n = 50 neuromasts (10 fish); Mtz: 21.11 ± 22.51, n = 50 neuromasts (10 fish); mean ± SD; Mann Whitney U test, p=0.0004.

DOI: https://doi.org/10.7554/eLife.43736.031

The following source data and figure supplements are available for figure 11:

**Source data 1.** DV cells are replenished by other support cell populations
DOI: https://doi.org/10.7554/eLife.43736.036

**Figure supplement 1.** Ablation of DV cells does not decrease the number of other support cell populations.
DOI: https://doi.org/10.7554/eLife.43736.032

**Figure supplement 1—source data 1.** Ablation of DV cells does not decrease the number of other support cell populations
DOI: https://doi.org/10.7554/eLife.43736.033

**Figure supplement 2.** Expression of nlsEos transgenes does not alter number of *sost*:NTR-GFP cells.
DOI: https://doi.org/10.7554/eLife.43736.034

**Figure supplement 2—source data 1.** Expression of nlsEos transgenes does not alter number of *sost*:NTR-GFP cells
DOI: https://doi.org/10.7554/eLife.43736.035

populations of support cells marked by transgene insertion. While our study may not definitively link the action of underlying loci with progenitor identity, our experiments demonstrate that these genetically labeled support cells have distinct progenitor functions, and can serve as important tools in future studies determining the precise mechanisms underlying regeneration in the lateral line.

## The role of Planar Cell Polarity and progenitor localization

Neuromasts located on the trunk develop at different times from different migrating primordia. Within a given neuromast, hair cells are arranged such that their apical stereocilia respond to

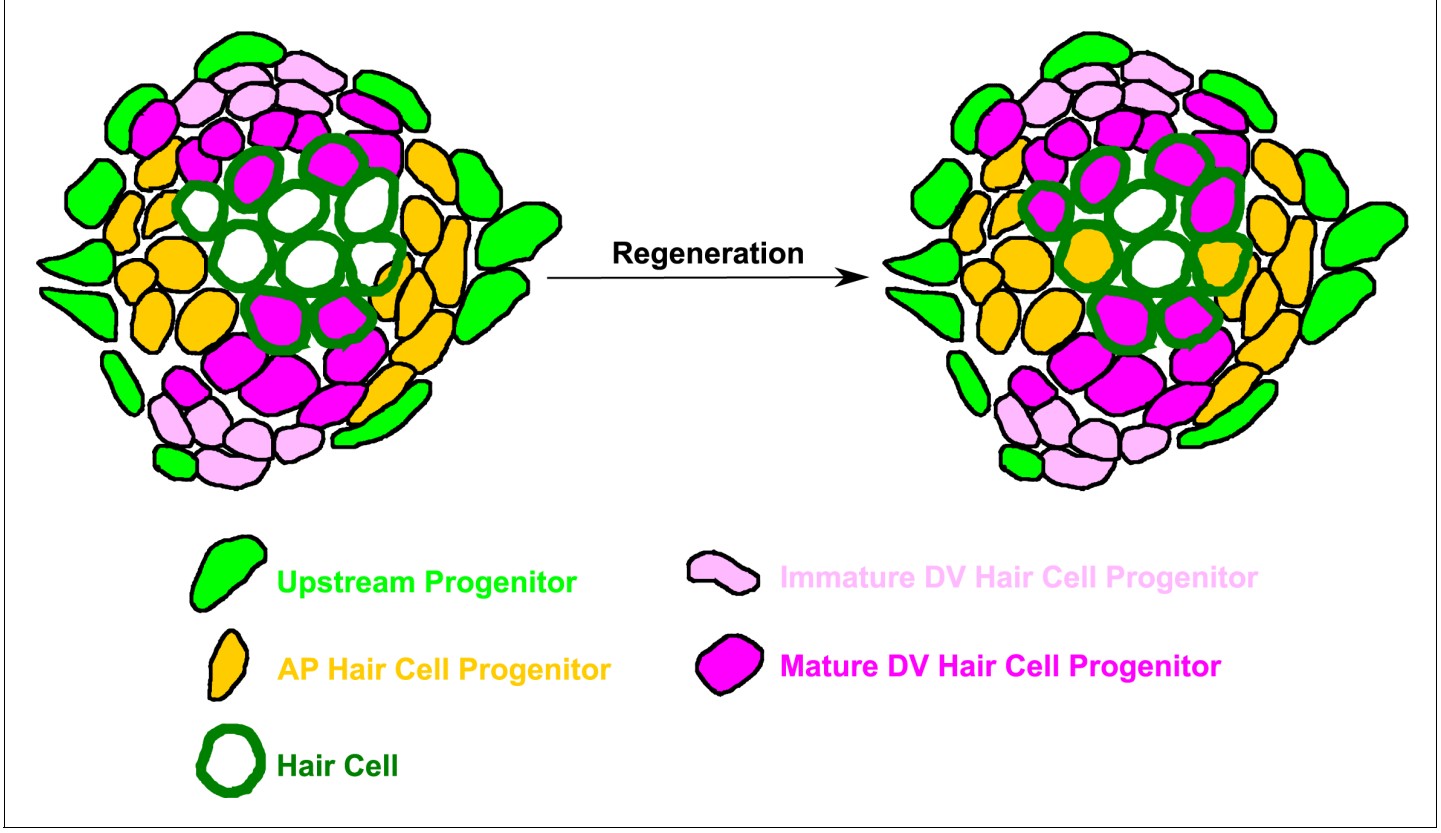

**Figure 12.** Model of neuromast progenitor identity. *Sost*:nlsEos-positive cells, located in the dorsoventral (DV) region of the neuromast, contain immature hair cell progenitors (shown in light pink) and mature hair cell progenitors (shown in magenta). Immature hair cell progenitors do not directly generate new hair cells (outlined in dark green) during regeneration, but do become mature hair cell progenitors, which comprise the majority of hair cell progenitors (see magenta-filled hair cells following regeneration). *Tnfsf10l3*:nlsEos-positive cells (shown in gold), located in the anteroposterior (AP) region of the neuromast, also serve as hair cell progenitors (see gold-filled hair cells following regeneration). Both of these populations are regulated by Notch signaling, and both can replenish immature hair cell progenitors. Finally, *sfrp1a*:nlsEos-positive cells (shown in light green), located in the periphery, do not serve as hair cell progenitors, nor are they regulated by Notch signaling. However, they are capable of replenishing immature hair cell progenitors, and can thus be classified as an upstream progenitor.

DOI: https://doi.org/10.7554/eLife.43736.037

directional deflection in one of two directions along the body axis. Hair cells derived from the first primordium (primI) respond along the anteroposterior axis, and hair cells derived from the second primordium (primII) respond along the dorsoventral axis (*López-Schier et al., 2004*; *López-Schier and Hudspeth, 2006*). Spatial restriction of support cell proliferation is orthogonal to hair cell planar polarity, with proliferation occurring dorsoventrally in primI-derived neuromasts and antero-posteriorly in primII-derived neuromasts (*Romero-Carvajal et al., 2015*). This 90-degree switch between prim1- and primII-derived neuromasts is reflected in the distribution of labeled cell populations as well: *tnfsf10l3*:nlsEos and *sost*:nlsEos retain their asymmetric localization in primII-derived neuromasts, but *tnfsf10l3*:nlsEos is predominantly expressed in the dorsoventral region and *sost*:nlsEos is restricted to the anteroposterior region (*Figure 2—figure supplement 1*). Thus, within a given neuromast along the trunk, expression of *sost*:nlsEos is orthogonal to hair cell planar polarity, and that of *tnfsf10l3*:nlsEos is parallel to hair cell planar polarity.

The relationship between asymmetric progenitor localization and hair cell planar polarity remains unknown. Planar cell polarity (PCP) signaling often drives asymmetry in other tissues and has been implicated in the planar polarity of lateral line hair cells. Zebrafish deficient in Vangl2, a critical component of the PCP pathway, still develop neuromasts, but their hair cells are oriented randomly toward one another and do not respond along a single axis (*López-Schier and Hudspeth, 2006*). Furthermore, this random orientation stems from misaligned divisions of hair cell progenitors

(*Mirkovic et al., 2012*). The transcription factor Emx2 has also been recently implicated in determining hair cell planar polarity (*Jiang et al., 2017*). Whether these genes or other components of PCP signaling mediate the asymmetric expression of *sost*:nlsEos and *tnfsf10l3*:nlsEos remains to be determined. It would also be interesting to examine whether hair cell planar polarity is influenced by the asymmetric localization of these progenitor populations, or vice versa.

## Regeneration of support cells in the absence of hair cell damage

Since zebrafish are able to properly regenerate their hair cells after multiple successive insults (*Cruz et al., 2015*; *Pinto-Teixeira et al., 2015*), and both daughters of progenitors give rise to hair cells, there must be a means of replenishing hair cell progenitors. Our EdU/BrdU double labeling experiment qualitatively demonstrated that hair cell progenitors could be replenished via proliferation of other support cells. It was thus unsurprising that DV cells could themselves regenerate. It is notable, however, that DV cells could be replenished even in the absence of hair cell damage, which means that hair cell death is not the sole signal that triggers support cell proliferation. Support cell regeneration in the absence of hair cell death has been observed in mammals, as certain types of cochlear support cells (inner border cells and inner phalangeal cells) are capable of regeneration following selective ablation, a process that occurs via transdifferentiation (*Mellado Lagarde et al., 2014*). In contrast, zebrafish DV cell regeneration primarily occurs via proliferation, as a majority of new DV cells were EdU-positive following Mtz-induced ablation. DV cells that were not EdU-positive could have arisen after the EdU pulse, been retained due to incomplete ablation, or potentially resulted from transdifferentiation from another source.

All three labeled support cell populations were able to replenish DV cells following Mtz-induced ablation. DV cells themselves contributed nearly 60% of new DV cells, although we cannot rule out that this number is inflated due to incomplete ablation. This result suggests that DV cells choose to either generate new hair cells or replenish lost DV cells, undergoing a form of self-renewal that does not require asymmetric division. The AP population is also capable of replenishing DV cells. That both of these populations can generate new DV cells is consistent with recent findings from *Viader-Llargués et al. (2018)*. This study defined support cells as a peripheral mantle population and a central sustentacular population. Following laser ablation of large portions of the neuromast, they found that sustentacular cells were able to regenerate mantle cells and other sustentacular cells, as well as hair cells, and could thus be considered tripotent progenitors. The transgenic lines they used to label sustentacular cells were broadly expressed, and thus should encompass both AP and DV populations. We were unfortunately unable to drive functional expression of NTR in Peripheral or AP cells, and were thus unable to test how ablation of these support cell populations would affect regeneration. However, we can conclude that the DV and AP populations are at least bipotent, since they can both generate new hair cells and new DV cells.

We found that the Peripheral population could also generate new DV cells following Mtz-induced ablation. Furthermore, it contributes more to DV regeneration than does the AP population. This was especially surprising since proliferation has rarely been observed in peripheral cells, at least during normal hair cell regeneration (*Ma et al., 2008*; *Romero-Carvajal et al., 2015*). However, the loss of a progenitor population could be considered to be a case of extreme damage to the neuromast, thus prompting the Peripheral cell population to respond. That Peripheral cells can serve as progenitors only in extreme circumstances is consistent with the findings of *Romero-Carvajal et al. (2015)* and *Viader-Llargués et al. (2018)*. Both studies suggested that mantle cells are capable of regenerating other cell types in the neuromast following extreme damage. However, the latter study found that mantle cells could only generate other mantle cells. Whether the Peripheral cell population in particular is capable of doing the same remains to be tested. Mantle cells have also been shown to proliferate following tail amputation, forming a migratory placode that forms new neuromasts along the regenerated tail (*Jones and Corwin, 1993*; *Dufourcq et al., 2006*). Given the differences across studies, we have hesitated to designate the *sfrp1a*:nlsEos labeled cells as mantle cells and have instead adopted the 'Peripheral' label. Since Peripheral cells can generate hair cell progenitors, we have characterized them as 'upstream progenitors' (*Figure 12*).

The zebrafish lateral line system continues to grow through larval and adult stages (*Nuñez et al., 2009*; *Ledent, 2002*; *Sapède et al., 2002*), with new neuromasts formed from budding from extant neuromasts (*Nuñez et al., 2009*; *Wada et al., 2013a*; *Wada et al., 2013b*) and generated anew from interneuromast cells, latent precursors deposited between neuromasts by the migrating

primordium (*Nuñez et al., 2009*; *Grant et al., 2005*). These interneuromast cells can also serve as progenitors in the event of whole neuromast ablation (*Sánchez et al., 2016*). We note that the *sfrp1a*:nlsEos transgene is expressed in interneuromast cells as well as Peripheral neuromast cells. Whether these cells share similar properties to generate new neuromasts remains to be tested.

Our model of neuromast progenitor identity does bear some similarities with other regenerative tissues. Both the hair follicle and intestinal epithelium contain a niche of stem cells (bulge cells and crypt cells, respectively) which generate transit-amplifying cells that are able to generate other cell types (*Ito et al., 2005*; *Taylor et al., 2000*; *Barker et al., 2007*). Due to their high rate of proliferation and multipotency, the DV cells in the neuromast could be likened to these transit-amplifying cells. However, progenitors in the neuromast may bear the most similarity to those of the olfactory epithelium, which contains two distinct progenitor populations: globose basal cells (GBCs), which are transit-amplifying cells that can restore lost olfactory neurons; and horizontal basal cells (HBCs), a quiescent population that can generate multiple cell types, including GBCs, in instances of extreme damage (*Iwai et al., 2008*; *Leung et al., 2007*). In our model, the DV and Peripheral cells are comparable to the GBCs and HBCs, respectively. However, we cannot make the claim that the Peripheral population is a resident stem cell population (like bulge cells, crypt cells, and HBCs), as we do not yet know if it is capable of self-renewal or of generating every cell type within the neuromast.

## Notch signaling differentially regulates support cell populations

Inhibition of Notch signaling during hair cell regeneration significantly increased the number of hair cells derived from both DV and AP cells, which was not unexpected given that both are hair cell progenitor populations. However, Notch inhibition had a greater impact on the AP population than on the DV population, suggesting that it may be more strongly regulated by Notch signaling. The receptor *notch3*, in particular, is most strongly expressed in the anteroposterior portions of the neuromast (*Wibowo et al., 2011*; *Romero-Carvajal et al., 2015*). Furthermore, a transgenic reporter of Notch activity is also expressed in the anteroposterior region (*Romero-Carvajal et al., 2015*; *Wibowo et al., 2011*). It is thus likely that asymmetrically-localized Notch signaling maintains quiescence among AP cells during homeostasis and is responsible for suppressing the contribution of the AP population to hair cell regeneration (compared to DV contribution). Since Notch signaling is not as strong in the dorsoventral regions of the neuromast, the DV population is less affected and could already be more 'primed' to serve as hair cell progenitors than the AP population. In addition, Notch signaling regulates Wnt signaling in the neuromast, particularly through activation of the Wnt inhibitor *dkk2* (*Romero-Carvajal et al., 2015*). It is possible that Notch signaling may repress Wnt signaling in AP cells, further contributing to their low regenerative capacity. More work needs to be done regarding the role of Wnt signaling in these support cell populations.

Notch inhibition did not have any impact on the Peripheral population's contribution to hair cell regeneration, indicating that Notch signaling does not suppress hair cell production by Peripheral cells. While it is difficult to tell from in situ expression, it seems that the Notch reporter is not active in the peripheral mantle cells (*Wibowo et al., 2011*; *Romero-Carvajal et al., 2015*). Thus, there must be some other mechanism, either intrinsic or extrinsic, that maintains relative quiescence among the Peripheral population.

It is not clear why these distinct populations of progenitors exist, as there is no clear difference in the types of hair cells they produce. Hair cells polarized in opposing direction are daughters of the final division of the hair cell progenitor (*López-Schier and Hudspeth, 2006*). Heterogeneity has also been recently described in the synaptic responses of lateral line hair cells (*Zhang et al., 2018*), but these differences appear lineage-independent. Instead, the allocation of distinct progenitors may serve an advantage with respect to their differential regulation. For example, our data suggest that DV cells contribute more to homeostatic addition of new hair cells to the neuromast in the absence of damage, while both AP and DV cells are engaged after hair cell damage. AP cells were unable to overcome the loss of the DV population, suggesting that feedback mechanisms regulating both hair cell production and progenitor replacement operate independently. While both AP and DV populations are regulated by Notch signaling, this regulation appears to operate independently as Notch inhibition does not compensate for DV ablation. Independent progenitors may offer the flexibility to add new hair cells under a variety of conditions for both ongoing hair cell turnover and in the face of catastrophic hair cell loss.

# Materials and methods

## Key resources table

| Reagent type (species) or resource | Designation | Source or reference | Identifiers | Additional information |
|---|---|---|---|---|
| Strain, strain background (*Danio rerio*) | *sfrp1a*:nlsEos | This paper | | Tg[*sfrp1a*:nlsEos]$^{w217}$; CRISPR-mediated transgenesis |
| Strain, strain background (*Danio rerio*) | *sfrp1a*:GFP | This paper | | Tg[*sfrp1a*:GFP]$^{w222}$; CRISPR-mediated transgenesis |
| Strain, strain background (*Danio rerio*) | tnfsf10l3:nlsEos | This paper | | Tg[*tnfsf10l3*:nlsEos]$^{w218}$; CRISPR-mediated transgenesis |
| Strain, strain background (*Danio rerio*) | tnfsf10l3:GFP | This paper | | Tg[*tnfsf10l3*:GFP]$^{w223}$; CRISPR-mediated transgenesis |
| Strain, strain background (*Danio rerio*) | *sost*:nlsEos | This paper | | Tg[*sost*:nlsEos]$^{w215}$; CRISPR-mediated transgenesis |
| Strain, strain background (*Danio rerio*) | *sost*:NTR-GFP | This paper | | Tg[*sost*:NTR-GFP]$^{w216}$; CRISPR-mediated transgenesis |
| Strain, strain background (*Danio rerio*) | myo6:mKate2 | This paper | | Tg[myosin6b:mKate2]$^{w232}$; Gateway cloning and Tol2-mediated transgenesis |
| Strain, strain background (*Danio rerio*) | brn3c:GFP | PMID: 15930106 | ZFIN ID: ZDB-TGCONSTRCT-070117–142 | Tg[Brn3c:GAP43-GFP]$^{s356t}$ |
| Strain, strain background (*Danio rerio*) | myo6:GFP | PMID: 27991862 | ZFIN ID: ZDB-TGCONSTRCT-170321–2 | Tg[myosin6b:GFP]$^{w186}$ |
| Antibody | anti-Parvalbumin (mouse monoclonal) | Millipore Sigma | MAB1572 | (1:500) |
| Antibody | anti-GFP (rabbit monoclonal) | Thermo Fisher Scientific | ABfinity G10362 | (1:500) |
| Antibody | anti-BrdU (mouse monoclonal) | Developmental Studies Hybridoma Bank | DHSB: G3G4; RRID: AB_2314035 | (1:100) |
| Recombinant DNA reagent | pBSK mbait-GFP | PMID: 25293390 | | |
| Recombinant DNA reagent | mbait-nlsEos | This paper | | pDEST vector expressing nuclear-localized Eos following the mbait-HSP sequence from pBSK mbait-GFP; see Materials and methods for more info. |
| Recombinant DNA reagent | pBSK mbait-NTR-GFP | This paper | | Modified pBSK mbait-GFP vector expressing enhanced potency nitroreductase fused to GFP; generated via Gibson assembly |
| peptide, recombinant protein | Cas9 protein with NLS (injection ready) | PNA Bio | PNA Bio: CP02 | |
| Commercial assay or kit | NucleoSpin Plasmid | Machery-Nagel | 740588.25 | |
| Commercial assay or kit | NucleoSpin Gel and PCR Clean-up | Machery-Nagel | 740609.25 | |
| Commercial assay or kit | Plasmid Maxi | Qiagen | 12163 | |
| Commercial assay or kit | RNA Clean and Concentrator | Zymo Research | RCC-25 | |

*Continued on next page*

*Continued*

| Reagent type (species) or resource | Designation | Source or reference | Identifiers | Additional information |
|---|---|---|---|---|
| Chemical compound, drug | Gateway BP Clonase II | Thermo Fisher Scientific | 11789020 | |
| Chemical compound, drug | Gateway LR Clonase II | Thermo Fisher Scientific | 11791020 | |
| Chemical compound, drug | 5x Isothermal Reaction Buffer | PMID: 19363495 | | |
| Chemical compound, drug | Metronidazole | Sigma-Aldrich | Sigma-Aldrich: M1547 | |
| Chemical compound, drug | LY411575 | Sigma-Aldrich | Sigma-Aldrich: SML0506 | |
| Chemical compound, drug | Neomycin | Sigma-Aldrich | Sigma-Aldrich: N1142 | |
| Chemical compound, drug | f-ara-EdU | Sigma-Aldrich | Sigma-Aldrich: T511293 | |
| Chemical compound, drug | BrdU | Sigma-Aldrich | Sigma-Aldrich: B5002 | |
| Chemical compound, drug | FM 1-43FX | Molecular Probes | Molecular Probes: F35355 | |
| Software, algorithm | GraphPad Prism | GraphPad Software | www.graphpad.com | |
| Software, algorithm | Slidebook | Intelligent Imaging Innovations (3i) | www.intellgent-imaging.com | |
| Software, algorithm | Zen Black | Zeiss | www.zeiss.com | |
| Software, algorithm | FIJI | PMID: 22743772 | | |

## Fish maintenance

Experiments were conducted on 5–8 dpf larval zebrafish (except for the double hair cell ablation experiment, which was conducted on 15–18 dpf fish). Larvae were raised in E3 embryo medium (14.97 mM NaCl, 500 µM KCL, 42 µM $Na_2HPO_4$, 150 µM $KH_2PO_4$, 1 mM $CaCl_2$ dihydrate, 1 mM $MgSO_4$, 0.714 mM $NaHCO_3$, pH 7.2) at 28.5°C. All wildtype animals were of the AB strain. Zebrafish experiments and husbandry followed standard protocols in accordance with University of Washington Institutional Animal Care and Use Committee guidelines.

## Plasmid construction

The myo6b:mKate2 construct was generated via the Gateway Tol2 system (Invitrogen). A pME-mKate2 (the mKate2 sequence being cloned from pMTB-Multibow-mfR, Addgene #60991) construct was generated via BP Recombination, and then a pDEST-myo6b:mKate2 construct was generated via LR recombination of p5E-myo6b, pME-mKate2, p3E-pA, and pDEST-iTol2-pA2 vectors. The mbait-GFP construct was a gift from Shin-Ichi Higashijima's lab. The mbait-nlsEos construct was also generated via Gateway LR recombination of p5E-mbait/HSP70l, pME-nlsEos, p3E-pA, and pDEST-iTol2-pA2 vectors. The mbait-epNTR-GFP construct was generated via Gibson assembly (*Gibson et al., 2009*), inserting the coding sequence of epNTR (cloned from pCS2-epNTR obtained from Harold Burgess' lab) plus a small linker sequence in front of the GFP in the original pBSK mbait-GFP vector. All plasmids were maxi prepped (Qiagen) prior to injection.

## CRISPR guides

Gene-specific guide RNA (gRNA) sequences were as follows:

*sfrp1a*: GTCTGGCCTAAAGAGACCAG
*tnfsf10l3*: GGGCTTGTATAGGAGTCACG
*sost*: GGGAGTGAGCAGGGATGCAAGGGCGAAGAACGGTGGAAGG

All gRNAs were synthesized according to the protocol outlined in *Shah et al. (2015)*, but were purified using a Zymo RNA Clean and Concentrator kit. Upon purification, gRNAs were diluted to 1 μg/μL, aliquoted into 4 μL aliquots, and stored at −80°C. *Sfrp1a* and *tnfsf10l3* gRNA sequences were designed via http://crispr.mit.edu, and the *sost* gRNA sequences were designed via http://crisprscan.org. The *tnfsf10l3* gRNA was targeted 388 base pairs upstream of the gene's start ATG codon, whereas the *sfrp1a* and *sost* guides were targeted to exons (the *sfrp1a* guide was targeted to exon 1, and the *sost* guides were targeted to exon2 and exon 1, respectively, in the order listed above).

## Tol2 transgenesis

The Tg[myosin6b:mKate2]$^{w232}$ (hereafter called myo6:mKate2) line was generated via Tol2 transgenesis. 1–2 nL of a 5 μL injection mix consisting of 20 ng/μL myo6b:mKate2 plasmid, 40 ng/μL transposase mRNA, and 0.2% phenol red were injected into single cell wildtype embryos. Larvae were screened for expression at three dpf and transgenic $F_0$ larvae were grown to adulthood. $F_0$ adults were outcrossed to wildtype fish, transgenic offspring were once again grown to adulthood, and the resulting adults were used to maintain a stable line.

## CRISPR-mediated transgenesis

All support cell transgenic lines were generated via CRISPR-mediated transgenesis, first described by *Kimura et al. (2014)*. This technique utilizes the targeted double strand breaks of the CRISPR/Cas9 system to insert a reporter construct into a gene of interest. In addition to a gene-specific gRNA and Cas9 protein, single cell-embryos are injected with a plasmid that contains a reporter (such as GFP or nlsEos) downstream of a minimal promoter (in this case a heat-shock promoter) and a 'bait' sequence that is not present in the zebrafish genome (we used the mbait sequence) in addition to another gRNA that targets the bait sequence. Following injection, Cas9 will recognize both the gene-specific gRNA as well as the bait gRNA, cleaving both the genomic DNA and linearizing the plasmid. This reporter plasmid can then be integrated into the genome via non-homologous end joining. These reporters can be targeted just upstream of the start codon of a gene of interest (about 200–600 base pairs) in order to co-opt the gene's cis-regulatory elements (the strategy we used for *tnfsf10l3* insertions) or can also be targeted directly to exons (as we did for *sfrp1a* and for *sost* insertions). Exonic insertions can thus be used to visualize potential loss of function mutations (*Ota et al., 2016*).

For most injections, a 5 μL injection mix was made consisting of 200 ng/μL gene-specific gRNA, 200 ng/μL mbait gRNA, 800 ng/μL Cas9 protein (PNA Bio #CP02), 20 ng/μL mbait-reporter plasmid, and 0.24% phenol red. The gRNAs and Cas9 protein were mixed together first, then heated at 37°C for 10 min, after which the other components were added. In the case of *sost*, in which two gRNAs were co-injected, each gRNA was added to the mix at a final concentration of 100 ng/μL (so 200 ng/μL of total guide-specific gRNA). When reconstituting the Cas9 protein, DTT was added to a final concentration of 1 mM DTT (per manufacturer's instructions). This is highly recommended to reduce needle clogging during the injection process. 1–2 nL of these injection mixes were injected into single cell wildtype embryos. Larvae were screened for expression at three dpf and transgenic $F_0$ larvae were grown to adulthood. $F_0$ adults were outcrossed to wildtype fish, transgenic offspring were once again grown to adulthood, and the resulting adults were used to maintain a stable line.

## Photoconversion

In order to photoconvert multiple nlsEos fish at once, larvae were transferred to a 60 × 15 mm petri dish and placed in a freezer box lined with aluminum foil. Then, an iLumen 8 UV flashlight (procured from Amazon) was placed over the dish and turned on for 15 min. Following the UV pulse, larvae were returned to standard petri dishes to await experimentation.

## Drug treatments

For all drug treatments, zebrafish larvae were placed in baskets in six well plates to facilitate transfer of larvae between media. Larvae were treated at five dpf unless otherwise noted. All wells contained 10 mL of drug, E3 embryo medium with the same effective % DMSO as the drug (for mock

treatments), or plain E3 embryo medium for washout. Following treatment, the fish were washed twice into fresh E3 embryo medium by moving the baskets into adjacent wells in the row, then washed a third time by transferring them into a 100 mm petri dish with fresh E3 medium. All drugs were diluted in E3 embryo medium. The drug treatment paradigms were as follows: for hair cell ablation, 400 µM neomycin (Sigma) for 30 min; for *sost*:NTR ablation, 10 mM metronidazole (Mtz; Sigma) with 1% DMSO; for Notch inhibition, 50 µM LY411575 (LY; Sigma; effective DMSO concentration of 0.5%) for 24 hr; for the *sost* ablation/Notch inhibition experiment (*Figure 7*): 10 mM Mtz/ 50 µM LY for 8 hr, then 50 µM LY for 16 hr.

For double hair cell ablation studies, larvae that were treated with neomycin were raised on a nursery in the UW fish facility beginning at seven dpf and then treated with 400 µM neomycin again at 15 dpf in standard petri dishes (10 days following the first neomycin treatment). These juvenile fish were washed into fresh system water multiple times before being returned to the nursery and were then fixed three days later (18 dpf).

## Vital Dye Labeling

Larvae were incubated in 3 µM FM 1-43FX (Molecular Probes) for one minute (consistent with *Owens et al., 2008* and *Stawicki et al., 2018*), then washed three times in fresh E3 embryo medium and immediately imaged. Incubation and washout were performed using the same manner as outlined above in the drug treatments section. Treatment occurred at either 5 dpf or 8 dpf.

## EdU and BrdU treatments

Following hair cell ablation with neomycin, larvae were incubated in 500 µM F-ara-EdU (Sigma #T511293) for 24 hr. Following *sost*:NTR ablation with Mtz, larvae were incubated in the same concentration of EdU for 48 hr. Larvae were placed into fresh EdU after the first 24 hr. F-ara-EdU was originally reconstituted in 50% $H_2O$ and 50% DMSO to 50 mM, so the working concentration of DMSO of 500 µM was 0.5%. In the case of the double ablation studies, juvenile fish were incubated in 10 mM BrdU (Sigma) with 1% DMSO in system water (used in the UW fish facility) following the second neomycin treatment for 24 hr. Following treatment, larvae were washed in fresh system water several times.

## Immunohistochemistry

Zebrafish larvae were fixed in 4% paraformaldehyde in PBS containing 4% sucrose for either 2 hr at room temperature or overnight at 4°C. Larvae were then washed three times (20 min each) in PBS containing 0.1% Tween20 (PBT), incubated for 30 min in distilled water, then incubated in antibody block (5% heat-inactivated goat serum in PBS containing 0.2% Triton, 1% DMSO, 0.02% sodium azide, and 0.2% BSA) for at least one hour at room temperature. Larvae were then incubated in mouse anti-parvalbumin or rabbit anti-GFP (or sometimes both simultaneously) diluted 1:500 in antibody block overnight at 4°C. The next day, larvae were once again washed three times (20 min each) in PBT, then incubated in a fluorescently-conjugated secondary antibody (Invitrogen, Alexa Fluor 488, 568, and/or 647) diluted 1:1000 in antibody block for 4–5 hr at room temperature. From this point onward, larvae were protected from light. Larvae were then rinsed three times (10 min each) in PBT and then stored in antibody block at 4°C until imaging. For BrdU immunohistochemistry, juvenile fish were rinsed once in 1N HCl, then incubated in 1N HCl following washout of Click-iT reaction mix (see below). IHC proceeded as above, except that the antibody block contained 10% goat serum and the fish were incubated in mouse anti-BrdU at a dilution of 1:100. All wash and incubation steps occurred with rocking.

## Click-iT

Cells that had incorporated F-ara-EdU were visualized via a Click-iT reaction. In the case of the double hair cell ablation experiment, Click-iT was performed before immunohistochemistry. Following fixation, fish were washed three times (10 min each) in PBT, then permeabilized in PBS containing 0.5% Triton-X for 30 min, then washed another three times (10 min each) in PBS. Next, fish were incubated for 1 hr at room temperature in a Click-iT reaction mix consisting of 2 mM $CuSO_4$, 10 µM Alexa Fluor 555 azide, and 20 mM sodium ascorbate in PBS (made fresh). Fish were protected from light from this point onwards. Afterwards, the standard IHC protocol listed above was performed

(beginning with the 3 20 min washes in PBT). For the *sost*:NTR regeneration experiment, the Click-iT reaction was performed after IHC. Following incubation in secondary antibody, larvae were washed three times (10 min each) in PBS, then incubated in 700 µL of the Click-iT reaction mix (again, made immediately prior to incubation) for 1 hr at room temperature. Larvae were then washed six times (20 min each) in PBT to ensure proper clearing of background labeling and stored in antibody block at 4°C until imaging.

## Confocal imaging

With the exception of imaging requiring a far-red laser (*Figure 1F–N*, *Figure 3C–I*, *Figure 5*) and *Figure 2—figure supplement 2A and D*, all imaging was performed using an inverted Marianas spinning disk system (Intelligent Imaging Innovations, 3i) with an Evolve 10 MHz EMCCD camera (Photometrics) and a Zeiss C-Apochromat 63x/1.2W numerical aperture water objective. For *Figure 2—figure supplement 2A and D*, larvae were imaged with a Zeiss Plan-Apochromat 20x/0.8 objective. All imaging experiments were conducted with fixed larvae ages 5–8 dpf. Fish were placed in a chambered borosilicate coverglass (Lab-Tek) containing 2.5–2.5 mL E3 embryo medium and oriented on their sides with a slice anchor harp (Harvard Instruments). Imaging was performed at ambient temperature, generally 25°C. Fish were positioned on their sides against the cover glass in order to image the first five primary neuromasts of the posterior lateral line (P1-P5). All imaging was performed with camera intensification of 650, gain of 3. Exposure time between 25–1500 ms was determined empirically to maintain imaging within the camera's linear range; imaging parameters were held constant across all groups within an experiment. Step size was 1 µm. All 3i Slidebook images were exported as .tif files to Fiji.

In cases when a far-red laser was required, imaging was performed on a Zeiss LSM 880 microscope with a Zeiss C-Apochromat 40x/1.2W numerical water objective. Fish were immersed in a solution of 50%glycerol/50% PBS, and then mounted on a plain microscope slide (Richard-Allen) beneath a triple wholemount coverslip. Imaging was performed at ambient temperature, generally 25°C. Fish were positioned on their sides against the cover glass in order to image the first five primary neuromasts of the posterior lateral line (P1-P5). For the double hair cell ablation experiment, following fixation the tails of fish were cut off and mounted underneath a single wholemount coverslip. The 3 neuromasts of the terminal cluster were imaged per tail. All imaging was performed at 4-5x digital zoom with master gain between 500–800 (as set in controlling software) for 488, 561, and 647 lasers, and a step size of 1 µm. All images were captured through the Zen Black software and opened in Fiji as .czi files.

## Statistical analysis

All statistical analyses were done with GraphPad Prism 6.0. The Mann Whitney U test was used for comparisons between two groups, whereas the Kruskal-Wallis test, with a Dunn's post-test, was used for comparisons between three or more groups. Statistical significance was set at $p=0.05$. All data presented in this paper are from individual, non-pooled experiments. However, the following data are representative of multiple experimental trials (number of trials in parentheses): *Figure 1B–D* (3); *Figure 2* (2); *Figure 3A–E* (3); *Figure 6* (2); *Figure 7* (2); *Figure 9* (2); *Figure 10* (2); *Figure 11D–I* (2).

## Acknowledgements

We would like to thank David White and the staff of the UW Fish Facility for animal care and maintenance, Ivan Cruz for providing the *sost* gRNA, Shin-Ichi Higashijima for the mbait-GFP vector, Harold Burgess for the pCS2-epNTR vector, Madeleine Hewitt for assistance with data analysis, and Sarah Pickett for critical reading of the manuscript.

## Additional information

### Funding

| Funder | Grant reference number | Author |
|---|---|---|
| National Institutes of Health | T32GM007270 | Eric D Thomas |
| National Institutes of Health | T32HD007183 | Eric D Thomas |
| National Institutes of Health | T32DC536115 | Eric D Thomas |
| National Institutes of Health | R21DC015110 | David W Raible |
| Hearing Health Foundation | HRP | David W Raible |
| Hamilton and Mildred Kellogg Trust | | David W Raible |
| The Whitcraft Family Gift | | David W Raible |

The funders had no role in study design, data collection and interpretation, or the decision to submit the work for publication.

### Author contributions
Eric D Thomas, Conceptualization, Formal analysis, Investigation, Methodology, Writing—original draft, Writing—review and editing; David W Raible, Conceptualization, Formal analysis, Supervision, Funding acquisition, Investigation, Methodology, Project administration, Writing—review and editing

### Author ORCIDs
Eric D Thomas (iD) http://orcid.org/0000-0003-4300-2893
David W Raible (iD) http://orcid.org/0000-0002-5342-5841

### Ethics
Animal experimentation: This study was performed in strict accordance with the recommendations in the Guide for the Care and Use of Laboratory Animals of the National Institutes of Health. All of the animals were handled according to approved institutional animal care and use committee (IACUC) protocol 2997-01 of the University of Washington.

### Decision letter and Author response
Decision letter https://doi.org/10.7554/eLife.43736.040
Author response https://doi.org/10.7554/eLife.43736.041

## Additional files

### Supplementary files
• Transparent reporting form
DOI: https://doi.org/10.7554/eLife.43736.038

### Data availability
All data generated or analysed during this study are included in the manuscript and supporting files.

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
