## [Decision Letter]

Thank you for submitting your article "Distinct progenitor populations mediate regeneration in the zebrafish lateral line." for consideration by *eLife*. Your article has been reviewed by three peer reviewers, and the evaluation has been overseen by a Reviewing Editor and Didier Stainier as the Senior Editor. The following individual involved in the review of your submission has agreed to reveal his identity: Bruce Riley (Reviewer #2).

The reviewers have discussed the reviews with one another and the Reviewing Editor has drafted this decision to help you prepare a revised submission.

This study identified and investigated three genetically distinct subpopulations of supporting cells in their ability to give rise to sensory hair cells in the zebrafish neuromast during homeostasis and regeneration. Specific transgenic lines were generated for these three populations of supporting cells, which allow lineage tracing under various conditions tested. Whereas the three populations of supporting cells behaved differently in their ability to give rise to hair cells and response to Notch signaling, the dorsal-ventral supporting cells were confirmed to have the highest propensity to give rise to sensory hair cells.

Essential revisions:

1) More details for CRISPR-mediated transgenic lines need to be provided in terms of methodology, guide RNA and targeted exons.

2) The expression of the transgenic lines needs to be compared to the expression of respective endogenous genes.

Reviewer #1:

CRISPR-mediated transgenesis: I have several questions regarding these lines:

What are the endogenous expression patterns of *sfrp1a, tnfsf1013*, and *sost* in zebrafish neuromasts, and how does endogenous expression compare to the transgenes? At a minimum, the authors should perform in situs or refer to expression profiles shown in other studies. Transgene expression may not reflect gene expression; for example, McNulty et al. (Gene Expression Patterns, 2012) did not detect sost expression in 5 day old lateral-line neuromasts.

Were duplicate experiments performed on the GFP expression lines to confirm that nuclear expression of Eos didn't disrupt normal cell function?

"*sfrp1a* and *sost* guides were targeted to exons" which exons? What controls were used to determine whether the insertions disrupted normal gene expression and/or function?

Did expression of two different transgenes driven by the same endogenous promotor influence expression of the other (i.e. do expression levels of sost:NTR-GFP + *sost*:nsl-Eos appear similar to sost:NTR-GFP alone)? It seems multiple genes driven by the same promotor may be influenced by the same gene regulatory mechanisms.

Subsection “AP and DV Cells Define Separate Progenitor Populations”, last paragraph, Figure 9E: If the sole reason the number of Eos-expressing hair cells increased is because the total number of hair cells decreased, why is the data expressed in this way? Is there a better method for demonstrating the AP progenitor contribution remains unchanged?

The authors speculate that both *sost*:NTR-GFP and *sost*:nls-Eos transgenes initiate expression at the same time, but nls-Eos is retained longer. Yet the data in Figure 5C suggests that a greater number of support cells consistently express *sost*:nls-Eos from 3 dpf. Have the authors confirmed that both transgenes express at the same time in younger ages?

That *sost*:NTR-GFP/Mtz mediated ablation results in incomplete ablation of DV support cells as defined by *sost*:nsl-Eos expression seems like an important caveat, especially when interpreting the data in Figure 9. Could it be that what the authors observe is simply due to a reduction in the overall number of support cells?

Subsection “The Role of Planar Cell Polarity and Progenitor Localization”, first paragraph (data not shown): Show the data.

Subsection “Notch Signaling Differentially Regulates Support Cell Populations”, first paragraph: What about the role of DV Wnt signaling and Notch/Wnt interactions? This seems like a glaring oversight given the Romero-Carvajal et al., 2015 study mentioned in this paragraph.

*Reviewer #2:*

1) In the subsection “Hair Cell Progenitors are Replenished via Proliferation of Other Support Cells”, and Figure 1 data are presented showing that at least some cells can incorporate both BrdU and EdU after two rounds of hair cell ablation. This is perhaps not surprising but has not been previously documented and is therefore important. However, there should be some attempt to quantify the findings. How many examples were observed vs. how many specimens were analyzed?

2) On a technical note the authors should explain in the text whether the 3 knockins disrupt gene function. These lineage tracers are described as "transgenes", a term that typically implies random integration e.g. facilitated by TOL2. Since these lines were created using CRISPR to target specific loci, I feel the term "knockin" might be more appropriate, and the functional status of targeted loci should be explained. In embryos carrying both *sost*:nlsEos and *sost:*NTR-Gfp, are these null for sost function, and does loss this affect development, maintenance or regeneration of neuromasts?

3) Most of the figures quantify the number of hair cells following ablation, but there should be some attempt to quantify changes in support cells populations, similar to data on *sost*:NTR-Gfp+ cells shown in Figure 11. The total number of nlsEos+ cells should already be available from the Figure 11 data set and can provide additional information about dynamic changes in the 3 support cell populations.

4) I am puzzled by some of the images in Figure 10. Figure 10C shows that the number of *sost*:NTR-Gfp+ cells declines by about half following Ntz treatment, as expected, but the image in Figure 10B appears to show a greater number of NTR-Gfp+ cells than the mock control in 10A. This is possibly a function of the greater magnification used in 10B. Please clarify.

*Reviewer #3:*

1) It is remarkable to be able to label three distinct groups of support cells. How the authors accomplished this feat is not clear and is suggested to be beyond the scope of this paper. However, there needs to be some clarification about choosing these three genes, and a brief description of how the lines were generated in the Results section. A few details are provided in the Materials and methods, but it is not clear what an 'mbait' vector is, and how difficult it was to insert the Eos or NTR genes. Additional information would clear up some of the mystery, i.e. why these three genes were targeted, and would be of general interest to the reader.

2) The authors state that the three groups are 'spatially' distinct. Yet a number of the AP cells appear to be in similar positions as the peripheral cells. The shapes and positions look almost identical in some images. If the peripheral and AP lines were crossed, would the number of cells be additive in the peripheral regions? Because of this overlap in position, it would be more accurate to call them genetically distinct populations with some spatial differences (leading to the names of peripheral versus AP as a convenient designation).

3) The NTR ablation experiments rely on the efficacy of the individual promoters. In the maximum projection images, particularly the AP line, some cells are really dim, while others are bright. Is there an age difference that could explain the range of nlsEos levels seen in this line? In addition, the Materials and methods state that different exposures (25-1500 ms) were used to obtain the images. Unless the expression is comparable, it is unclear whether the experimental outcomes using NTR are due to the strength (or peak expression during development) of the individual promoters, or rather the roles of the different cell types in regeneration. How well does ablation work with the AP cell type and why doesn't it work at all with peripheral cells as stated in the discussion? Some clarification is needed.

4) The reticence of the peripheral cells is interesting. Did the authors try to ablate both AP and DV populations to see if this would result in more participation by the peripheral cells?

---

## [Author Response]

Essential revisions:1) More details for CRISPR-mediated transgenic lines need to be provided in terms of methodology, guide RNA and targeted exons.

More details about the selection of the specific genes have been added to the Results section (subsection “Different Progenitor Identities Among Distinct Support Cell Populations”, first paragraph), and additional information has been added to the Materials and methods section (subsection “CRISPR-mediated Transgenesis”, first paragraph).

2) The expression of the transgenic lines needs to be compared to the expression of respective endogenous genes.

A comprehensive description of gene expression in zebrafish neuromasts was just published by the Piotrowski lab (Lush et al., 2019). Expression of our transgenes largely matches those described. We include a discussion of this point (subsection “Differences in Hair Cell Progenitor Identity Among Support Cell Populations”, last paragraph), and address this point more fully below in response to reviewer 1.

Reviewer #1:

CRISPR-mediated transgenesis: I have several questions regarding these lines:What are the endogenous expression patterns of sfrp1a, tnfsf1013, and sost in zebrafish neuromasts, and how does endogenous expression compare to the transgenes? At a minimum, the authors should perform in situs or refer to expression profiles shown in other studies. Transgene expression may not reflect gene expression; for example, McNulty et al. (Gene Expression Patterns, 2012) did not detect sost expression in 5 day old lateral-line neuromasts.

As noted above, we now compare the expression of our transgenes to the recently published expression patterns found in Lush et al., 2019. However, we do not believe that whether our transgenes match the endogenous expression of the genes in question matters to the results of our study. Rather we are using transgene expression as tools for marking cell populations, so only that pattern of their expression is important. We have added a section in the Discussion addressing this point (subsection “Differences in Hair Cell Progenitor Identity Among Support Cell Populations”, last paragraph).

Were duplicate experiments performed on the GFP expression lines to confirm that nuclear expression of Eos didn't disrupt normal cell function?

While no experiments were done on the GFP lines, we include new figures supporting the idea that nlsEos expression does not alter proper hair cell development or regeneration alone (Figure 2—figure supplement 3) or in combination with NTR-GFP (Figure 6—figure supplement 2). We used FM 1-43FX dye to label hair cells in these experiments. As FM 1-43FX labeling serves as a proxy for hair cell activity, these experiments also indicate that hair cell function is unaffected by nlsEos expression. These transgenes also had no effect on the number of DV cells as assessed by NTR-GFP expression (Figure 11—figure supplement 2). Together, these experiments strongly suggest that transgene expression does not affect the processes we are studying.

"sfrp1a and sost guides were targeted to exons" which exons? What controls were used to determine whether the insertions disrupted normal gene expression and/or function?

The exons to which the exon-targeted guides were targeted are listed in the Materials and methods section (subsection “CRISPR Guides”). As noted above, transgene expression did not disrupt hair cell development or regeneration, or hair cell function as measured by FM 1-43FX dye.

Did expression of two different transgenes driven by the same endogenous promotor influence expression of the other (i.e. do expression levels of sost:NTR-GFP + sost:nsl-Eos appear similar to sost:NTR-GFP alone)? It seems multiple genes driven by the same promotor may be influenced by the same gene regulatory mechanisms.

We tested this idea by comparing the number of cells labeled by *sost*:NTR-GFP in the background of each nlsEos transgene (Figure 11—figure supplement 2),and found that expression of different support cell transgenes did not alter the number of cells expressing *sost*:NTR-GFP. The number of *sost*:NTR-GFP cells is the same in the *sost*:nlsEos background as in other support cell backgrounds, so this dual expression does not seem to impact expression.

Subsection “AP and DV Cells Define Separate Progenitor Populations”, last paragraph, Figure 9E: If the sole reason the number of Eos-expressing hair cells increased is because the total number of hair cells decreased, why is the data expressed in this way? Is there a better method for demonstrating the AP progenitor contribution remains unchanged?

We apologize for the confusion. The number of *tnfsf10l3*:nlsEos+ hair cells remained unchanged, as noted in the Results and quantified in Figure 9D. We were simply noting that the percentage of hair cells labeled by *tnfsf10l3*:nlsEos had increased, but this was only because the total number of hair cells decreased (both of which are referenced in the Results and quantified in Figure 9). We have clarified the language in that section to avoid confusion.

The authors speculate that both sost:NTR-GFP and sost:nls-Eos transgenes initiate expression at the same time, but nls-Eos is retained longer. Yet the data in Figure 5C suggests that a greater number of support cells consistently express sost:nls-Eos from 3 dpf. Have the authors confirmed that both transgenes express at the same time in younger ages?

We observe that both transgenes begin to be expressed around late 2 dpf/early 3 dpf, but we have not performed detailed studies to examine whether each comes on at exactly the same time. However, even if it is the case that one transgene is expressed slightly earlier than the other, that doesn’t change our findings that the overlap between the two changes substantially between 3 and 5 dpf (as shown in Figure 5). Thus, we believe our model of maturation from *sost*:NTR-GFP/nlsEos to just *sost*:nlsEos remains valid.

That sost:NTR-GFP/Mtz mediated ablation results in incomplete ablation of DV support cells as defined by sost:nsl-Eos expression seems like an important caveat, especially when interpreting the data in Figure 9. Could it be that what the authors observe is simply due to a reduction in the overall number of support cells?

We agree that incomplete ablation is an important caveat, one which we addressed in the Discussion (subsection “Regeneration of Support Cells in the Absence of Hair Cell Damage”, second paragraph). To test whether ablation of *sost*:NTR-GFP cells reduces other support cells, we examined if other support cell types were lost due to this ablation (Figure 11—figure supplement 1), and found no changes. Thus, our observation in Figure 9 is not simply due to an overall loss of *tnfsf10l3*+ support cells.

Subsection “The Role of Planar Cell Polarity and Progenitor Localization”, first paragraph (data not shown): Show the data.

Done. See Figure 2—figure supplement 1.

Subsection “Notch Signaling Differentially Regulates Support Cell Populations”, first paragraph: What about the role of DV Wnt signaling and Notch/Wnt interactions? This seems like a glaring oversight given the Romero-Carvajal et al., 2015 study mentioned in this paragraph.

We agree, and have added some discussion about the role of Wnt and Notch in the Discussion (subsection “Notch Signaling Differentially Regulates Support Cell Populations”, first paragraph).

Reviewer #2:

1) In the subsection “Hair Cell Progenitors are Replenished via Proliferation of Other Support Cells”, and Figure 1 data are presented showing that at least some cells can incorporate both BrdU and EdU after two rounds of hair cell ablation. This is perhaps not surprising but has not been previously documented and is therefore important. However, there should be some attempt to quantify the findings. How many examples were observed vs. how many specimens were analyzed?

Quantification has been added to the Results section.

2) On a technical note the authors should explain in the text whether the 3 knockins disrupt gene function. These lineage tracers are described as "transgenes", a term that typically implies random integration e.g. facilitated by TOL2. Since these lines were created using CRISPR to target specific loci, I feel the term "knockin" might be more appropriate, and the functional status of targeted loci should be explained. In embryos carrying both sost:nlsEos and sost:NTR-Gfp, are these null for sost function, and does loss this affect development, maintenance or regeneration of neuromasts?

While the method is different than for generating transgenics with Tol2, since the reporter construct includes a minimal heat-shock promoter and is not introduced by homologous recombination, we believe the term transgene is still appropriate.

We have added results that show that embryos positive for both *sost*:NTR-GFP and *sost*:nlsEo, have the same number of functional hair cells following development and regeneration as transgenic heterozygotes (i.e. *sost*:NTR-GFP alone and *sost*:nlsEos alone) as well as non-transgenic siblings (Figure 6—figure supplement 2). However, we did not explicitly test whether these double-positive fish were truly null for *sost*. As we note above, the aim of our study was not to examine the role of *sost* in hair cell development or regeneration, but to use the transgenes to mark cell fate.

3) Most of the figures quantify the number of hair cells following ablation, but there should be some attempt to quantify changes in support cells populations, similar to data on sost:NTR-Gfp+ cells shown in Figure 11. The total number of nlsEos+ cells should already be available from the Figure 11 data set and can provide additional information about dynamic changes in the 3 support cell populations.

We have added these data. As seen in Figure 11—figure supplement 1, there is no change in labeled cells in the *tnfsf10l3* or *sfrp1a* populations. There is the predicted decrease in converted cells in the *sost*:nlsEos line.

4) I am puzzled by some of the images in Figure 10. Figure 10C shows that the number of sost:NTR-Gfp+ cells declines by about half following Ntz treatment, as expected, but the image in Figure 10B appears to show a greater number of NTR-Gfp+ cells than the mock control in 10A. This is possibly a function of the greater magnification used in 10B. Please clarify.

The image in question was on the higher end of the spectrum in terms of number of NTR-GFP cells present following regeneration. The figure has been amended with images that are better representative of the data.

Reviewer #3:

1) It is remarkable to be able to label three distinct groups of support cells. How the authors accomplished this feat is not clear and is suggested to be beyond the scope of this paper. However, there needs to be some clarification about choosing these three genes, and a brief description of how the lines were generated in the Results section. A few details are provided in the Materials and methods, but it is not clear what an 'mbait' vector is, and how difficult it was to insert the Eos or NTR genes. Additional information would clear up some of the mystery, i.e. why these three genes were targeted, and would be of general interest to the reader.

As noted above, we include an expanded explanation of how CRISPR-mediated transgenesis works in the Materials and methods section and have clarified why these genes were selected in the Results section.

2) The authors state that the three groups are 'spatially' distinct. Yet a number of the AP cells appear to be in similar positions as the peripheral cells. The shapes and positions look almost identical in some images. If the peripheral and AP lines were crossed, would the number of cells be additive in the peripheral regions? Because of this overlap in position, it would be more accurate to call them genetically distinct populations with some spatial differences (leading to the names of peripheral versus AP as a convenient designation).

This is a fair point. We have modified the language in the paper to reflect this.

3) The NTR ablation experiments rely on the efficacy of the individual promoters. In the maximum projection images, particularly the AP line, some cells are really dim, while others are bright. Is there an age difference that could explain the range of nlsEos levels seen in this line? In addition, the Materials and methods state that different exposures (25-1500 ms) were used to obtain the images. Unless the expression is comparable, it is unclear whether the experimental outcomes using NTR are due to the strength (or peak expression during development) of the individual promoters, or rather the roles of the different cell types in regeneration. How well does ablation work with the AP cell type and why doesn't it work at all with peripheral cells as stated in the discussion? Some clarification is needed.

We ensured that all fish within a given clutch were at the same stage (i.e. 50% epiboly) when collecting them, so age shouldn’t be a factor. It could be that the *tnfsf10l3* gene is expressed at different levels at different ages, but that was not a focus of the study. Regarding the range of exposures listed, that was done to encompass all of the experiments performed, as some transgenes were brighter than others. However, exposure times for a given transgene were held constant across groups within a given experiment. Regarding the ablation of NTR cells, we unfortunately were unable to generate lines in which the Peripheral or AP populations could be ablated, as we were unable to knock-in functional NTR-GFP construct into either loci. Thus, we were unable to test whether DV or AP cells could replenish AP cells. That part of the discussion has been clarified to avoid confusion.

4) The reticence of the peripheral cells is interesting. Did the authors try to ablate both AP and DV populations to see if this would result in more participation by the peripheral cells?

We agree that it is interesting, and were hoping to do just that! However, as mentioned above, we were unable to generate a *tnfsf10l3*:NTR line, and were thus unable to ablate AP cells.